# Aryl amino acetamides prevent *Plasmodium falciparum* ring development via targeting the lipid-transfer protein *Pf*START1

Madeline G. Dans[1,2,3,4,13] ✉, Coralie Boulet [1,12,13], Gabrielle M. Watson[2,4], William Nguyen[2,4], Jerzy M. Dziekan[2,4], Cindy Evelyn [2,4], Kitsanapong Reaksudsan[2,4], Somya Mehra[1,3], Zahra Razook[1,3], Niall D. Geoghegan [2,4], Michael J. Mlodzianoski [2,4], Christopher Dean Goodman[5], Dawson B. Ling [1], Thorey K. Jonsdottir [1,6,7,8], Joshua Tong[2], Mufuliat Toyin Famodimu[9], Mojca Kristan [10], Harry Pollard[10], Lindsay B. Stewart[10], Luke Brandner-Garrod[10], Colin J. Sutherland [9,10], Michael J. Delves [9], Geoffrey I. McFadden[5], Alyssa E. Barry [1,3], Brendan S. Crabb[1,6,11], Tania F. de Koning-Ward [3], Kelly L. Rogers [2,4], Alan F. Cowman [2,4], Wai-Hong Tham [2,4], Brad E. Sleebs [2,4] & Paul R. Gilson [1,6] ✉

With resistance to most antimalarials increasing, it is imperative that new drugs are developed. We previously identified an aryl acetamide compound, MMV006833 (M-833), that inhibited the ring-stage development of newly invaded merozoites. Here, we select parasites resistant to M-833 and identify mutations in the START lipid transfer protein (PF3D7_0104200, *Pf*START1). Introducing *Pf*START1 mutations into wildtype parasites reproduces resistance to M-833 as well as to more potent analogues. *Pf*START1 binding to the analogues is validated using organic solvent-based Proteome Integral Solubility Alteration (Solvent PISA) assays. Imaging of invading merozoites shows the inhibitors prevent the development of ring-stage parasites potentially by inhibiting the expansion of the encasing parasitophorous vacuole membrane. The *Pf*START1-targeting compounds also block transmission to mosquitoes and with multiple stages of the parasite's lifecycle being affected, *Pf*START1 represents a drug target with a new mechanism of action.

There were an estimated 249 million cases of malaria in 2022 resulting in approximately 608,000 deaths, mainly in sub-Saharan Africa[1]. Artemisinin combination therapies (ACTs) remain the frontline treatments for malaria infections by eliminating the causative *Plasmodium* parasites from patients. Of concern is that mutations in the parasite's *kelch13* gene, a marker of partial resistance to ACTs, have now emerged in many regions around the world and portents to increasing treatment failures[2–6]. For this reason, new small molecule inhibitors with novel mechanisms of action that can kill various lifecycle stages of the parasite need to enter the developmental pipeline.

A prime process to target is invasion by the free-form merozoite into host red blood cells (RBCs)[7]. This process occurs rapidly in less than a minute[8,9] and requires many protein-protein interactions to successfully occur in a precise, carefully choreographed order[10]. To identify inhibitors with novel mechanisms of action against RBC invasion, we previously performed a phenotypic screen of the 400 compound Medicines for Malaria Venture (MMV) Pathogen Box[11]. One

hit compound, MMV006833 (M-833), had a novel effect whereby the merozoite entered its target RBC but failed to differentiate into an amoeboid ring-stage parasite during the formation of the parasitophorous vacuole membrane (PVM) around the parasite[11]. Here, we generated M-833 resistant parasites and whole genome sequencing revealed mutations in the *Plasmodium falciparum* Steroidogenic Acute Regulatory protein-related lipid Transfer (START) domain-containing phospholipid transfer protein (PF3D7_0104200, PFA0210c, also called PV6[12]).

START-domain proteins can ferry lipids between membranes and are defined by their START region of ~210 amino acids that form a hydrophobic lipid-binding pocket. START domains typically comprise an α/β helix-grip fold with antiparallel β-sheets, flanked by N- and C-terminal α-helices[13]. The human START domain family encompasses 15 members[14], whilst *Plasmodium spp*. in contrast has only five known START-domain proteins[15]. Of those five, PFA0210c is the most studied and the focus of this article, which we will refer to as *Pf*START1. *Pf*START1 shares sequence similarities of 20% with the human StarD7, which is a phosphatidylcholine transfer protein involved in maintaining mitochondrial membrane integrity. However, *Pf*START1 is structurally closer to human StarD2, another phosphatidylcholine transfer protein involved in lipid droplet metabolism[15,16]. In parasites, *Pf*START1 is essential during the blood stages and most strongly expressed during schizogony[16]. *Pf*START1 can transfer a wide range of phospholipids in vitro, with a preference for phosphatidylcholine and phosphatidylinositol[16] which is regulated by an unusual C-terminal extension that is essential for growth in vivo[17]. Due to its function and expression profile, previous studies have suggested that *Pf*START1 may play a role in forming the PVM upon merozoite invasion by transferring lipids from the RBC or parasite membranes into the nascent PVM, and therefore represents a promising drug target[16,17].

Here, we report the M-833 series as inhibitors of *Pf*START1. In this study, we identify mutations in the START domain of *Pf*START1 in parasites resistant to M-833. The introduction of these mutations into 3D7 wildtype parasites conferred resistance to M-833 and its highly potent analogues. We further investigate how the M-833 series binds to *Pf*START1 and how this blocks merozoite development into ring-stage parasites.

## Results

### Selection for resistance to MMV006833 followed by whole genome sequencing identified *Pf*START1 as a possible target

To select for resistance to MMV006833 (Fig. 1A) hereafter called M-833, five populations (A-E) of $10^8$ asexual blood stage 3D7 parasites were cycled on and off 3 μM of M-833 (corresponding to 10 x $EC_{50}$). After three cycles of drug treatment, three parasite populations were identified as less susceptible to the compound and their growth in serially diluted M-833 after 72 h was measured by lactate dehydrogenase (LDH) activity assays. The $EC_{50}$ of the drug-treated parasites was 2.18 to >10-fold higher than 3D7 wild-type parasites, demonstrating resistance had been generated (Fig. S1A). The two most resistant populations (PopD and PopE) were further cloned out by limiting dilution. Growth assays demonstrated the resistance against M-833 was heritable, with a 17-fold increase in $EC_{50}$ compared to 3D7 parasites (Fig. 1B). To identify the target of M-833, genomic DNA was extracted from the clonal lines of PopD and PopE and subjected to whole genome sequencing using MinION technology and compared to clonal 3D7 parental parasites[18]. This identified 14 non-synonymous single nucleotide polymorphisms (SNP) across nine different genes when compared to the 3D7 parental strain (Supplementary Data 1). The two genes that had SNPs present across all resistant clones were the *P. falciparum liver stage antigen 1* gene (PF3D7_1036400) and *P. falciparum stAR-related lipid transfer protein* (PF3D7_0104200, PFA0210c;

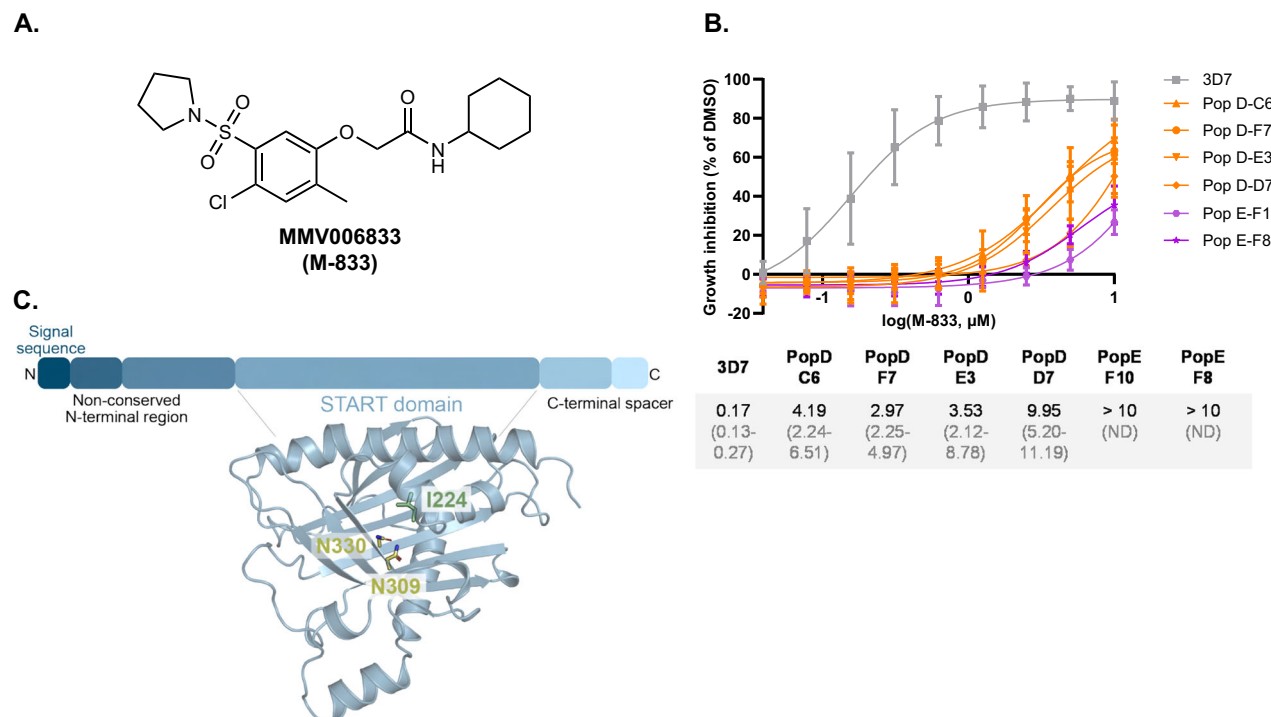

**A.**

**MMV006833 (M-833)**

**C.**

**B.**

| 3D7 | PopD C6 | PopD F7 | PopD E3 | PopD D7 | PopE F10 | PopE F8 |
|---|---|---|---|---|---|---|
| 0.17 (0.13-0.27) | 4.19 (2.24-6.51) | 2.97 (2.25-4.97) | 3.53 (2.12-8.78) | 9.95 (5.20-11.19) | > 10 (ND) | > 10 (ND) |

**Fig. 1 | M-833 resistant parasites contain mutations in the *P. falciparum* START1 protein. A** Structure of MMV006833 (M-833). **B** Parasites populations (Pop) resistant to M-833 were generated by exposing 3D7 *P. falciparum* parasites to 10 x $EC_{50}$ of M-833 three times. These parasites were cloned, and growth inhibition assays performed in the presence of 10 μM to 0.04 μM M-833. Growth inhibition was normalised to DMSO controls. Average of $EC_{50}$ values and 95% confidence intervals are indicated; *n* = 3 biological replicates. Source data are provided as a Source Data file. **C** Key domains and cartoon representation of the AlphaFold predicted structure of the StAR-related lipid transfer protein, *Pf*START1 (PF3D7_0104200)[70,71]. Mutations that had been identified in the M-833 resistant parasite populations are shown as sticks (yellow = N309K, N330K; green = I224).

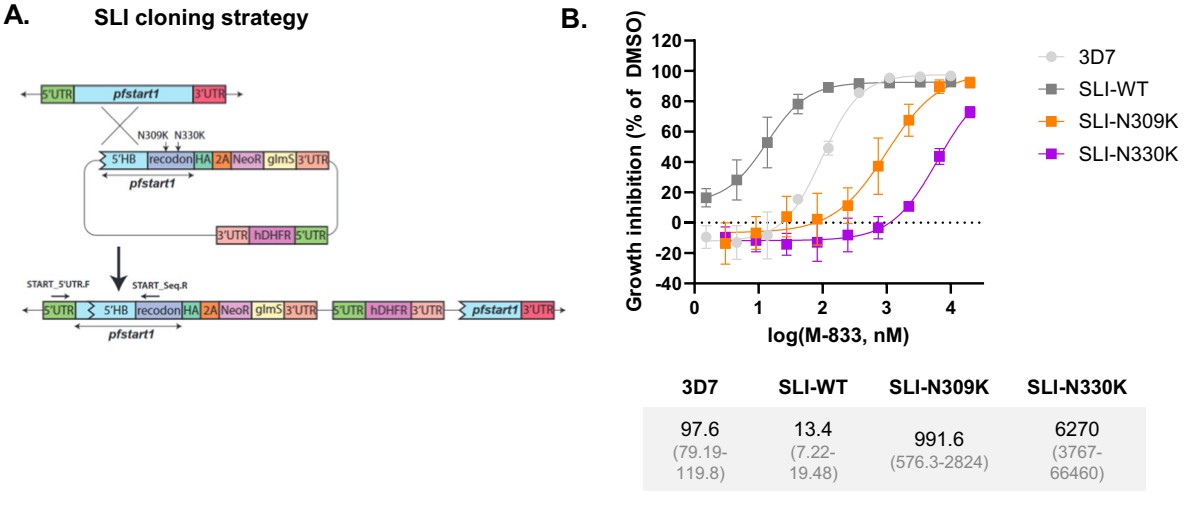

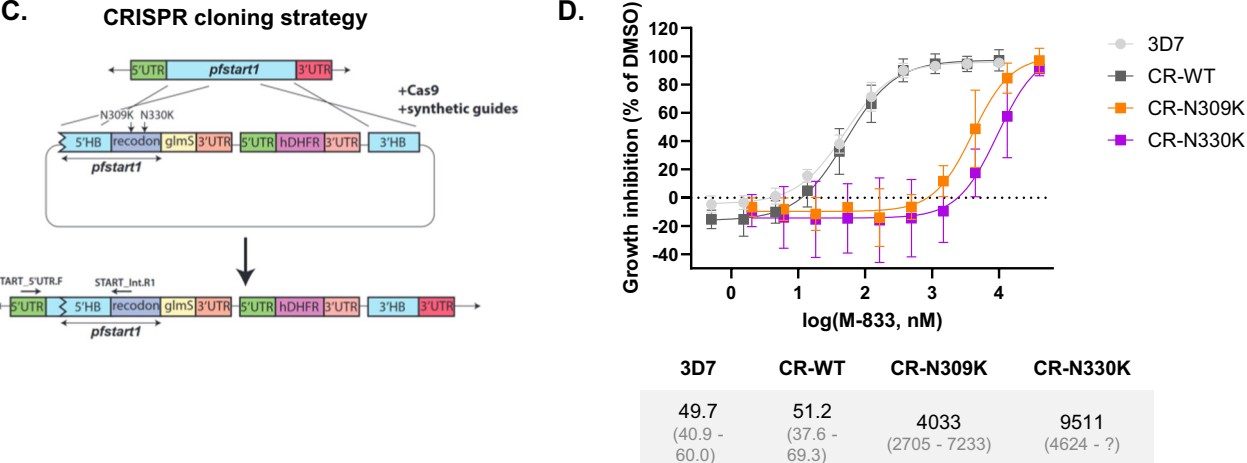

**Fig. 2 | Engineering mutations N309K and N330K in *Pf*START1 confers resistance to M-833.** Wild-type (WT) or mutant (N309K, N330K) *pfstart1* genes were introduced in drug-sensitive 3D7 parasites using a Selection-Linked Integration (SLI) method or using CRISPR-Cas9 (CR). **A** Schematic of the SLI cloning strategy. The 5' arm of native *pfstart1* was amplified, PCR sewn to a recodonised version of the 3' arm (either wildtype (WT), N309K or N330K), inserted into a p-HA-2A-Neo-glmS plasmid and transfected into 3D7 parasites, to be integrated into the native gene. **B** Growth inhibition curves of M-833 treated 3D7 and SLI-parasites. Parasites were incubated in various concentrations of M-833 and growth was assessed 72 h later using a lactate dehydrogenase activity assay. $EC_{50}$ values and 95% confidence intervals are indicated below the graph; $n = 3$ biological replicates, mean +/− SD. **C** Schematic of the CRISPR cloning strategy. The full length 5' homology blocks for WT, N309K, and N330K were amplified from the SLI plasmids and inserted into the p1.2-glmS[86] plasmid, along with the native 3' arm of *pfstart1*. Note that this construct does not include a HA tag. The plasmids were transfected into 3D7 parasites with Cas9 enzyme complexed with a guide RNA, to be integrated into the native gene. **(D)** Growth inhibition curves of M-833 on 3D7 and CRISPR-parasites (same as **B**). $n = 3$ biological replicates, mean +/− SD. Source data are provided as a Source Data file.

*pfstart1*). The former is not expressed in the blood stage[19–21] and comprises a large central repeat region which is problematic for variant calling. Therefore, we focused on *pfstart1*: two of the three sequenced PopD clones (PopD-C6 and -F7) had a SNP (AAC-AAG) which caused a mutation N309K in *Pf*START1 (Fig. 1C, Supplementary Data 1). The third PopD clone (PopD-E3) also contained the N309K mutation, although this had been removed during quality filtration as it did not meet the minimum depth of 10x coverage. While we did not sequence the genome of PopD-D7, it also contained the N309K mutation, as demonstrated by PCR amplification and sequencing (Fig. S1B). Both PopE clone genomes had a SNP (AAC-AAA) within *pfstart1* that resulted in a mutation N330K (Fig. 1C, Supplementary Data 1). Finally, PCR-amplification and sequencing of *pfstart1* of PopC showed it contained an I224F mutation (Fig. S1C). Collectively, all the identified mutations (N309K, I224F, N330K) were found within the central

conserved START domain and predicted to face the lipid-binding pocket (Fig. 1C).

**Introduction of N309K and N330K mutations into *pfstart1* gene increases resistance to M-833**

To test whether *Pf*START1 was the target of the M-833 series, the N309K and N330K mutations were first introduced into the *pfstart1* gene using selection-linked integration (SLI) that integrates via a single crossover homologous recombination[22]. This construct contained a haemagglutinin (HA) tag at the C-terminus and a *glmS* riboswitch to allow protein knock-down after addition of glucosamine (GlcN) which was separated by a P2A-skip peptide and neomycin resistance gene (Fig. 2A). After clonal lines of the WT, N309K and N330K had been obtained, integration was confirmed by PCR (Fig. S2A), and the modified *pfstart1* loci were sequenced to ensure the mutations were correct

(Fig. S2B). The clonal parasites obtained with the SLI method are hereafter called SLI-WT, SLI-N309K, and SLI-N330K.

We challenged SLI-WT, SLI-N309K and SLI-N330K with M-833 to determine the effect of the mutations on M-833 treatment (Fig. 2B). SLI-N309K and SLI-N330K parasites were 10- and 64-fold more resistant than the 3D7 parental parasites respectively. Unexpectedly, we found that the SLI-WT parasites, in which the recodonised WT *pfstart1* sequence was used to replace endogenous coding sequence, were 7-fold more sensitive to M-833 compared to the parental 3D7 parasites.

Since the addition of the HA tag and P2A skip peptide could interfere with *Pf*START1 function and sensitivity of SLI-WT parasites to M-833 a different tagging strategy was used, whereby the introduction of the mutations into *pfstart1* did not include HA and P2A, using CRISPR-Cas9[23]. In comparison to the SLI plasmids, the CRISPR constructs contained an additional 3' homology block designed to enable double crossover and limited plasmid insertion (Fig. 2C). Correct integration was confirmed by PCR (Fig. S2D), and the modified *pfstart1* loci were sequenced (Fig. S2E). These CRISPR parasite clones were hereafter called CR-WT, CR-N309K, and CR-N330K. CR-N309K and CR-N330K parasites exhibited resistance to M-833 to a similar extent as the original resistant parasites (over 81-fold resistance compared to CR-WT, Fig. 2D). In addition, the CR-WT parasites displayed an $EC_{50}$ against M-833 comparable to 3D7 (51.2 and 49.7 nM respectively). Overall, these results demonstrate that the mutations N309K and N330K in *Pf*START1 confer resistance to M-833.

### Knockdown of *Pf*START1 sensitises parasites to M-833

To examine if knocking down *Pf*START1 would decrease the $EC_{50}$ of M-833 due to a reduction in target protein levels we tested the knockdown efficiency exerted by the *glm*S riboswitch integrated downstream of the *pfstart1* genes (Fig. 3A, D). Wildtype plasmepsin V processed *Pf*START1 is predicted to be about 48 kDa as observed by western blot for 3D7 parasites (Fig. 3A). The 3xHA tag and P2A peptide add 5.8 kDa to the protein in line with that observed for the SLI-tagged parasites (Fig. 3A). In SLI-N309K, there was a smaller band detected that was similar in size to the native protein and contained no HA-tag. To remove any contaminating wildtype parasites the parasites were re-cloned, but all lines still contained the two *Pf*START1 bands. Diagnostic PCRs of the SLI-N309K parasites indicated the native gene was tagged so perhaps the smaller band arose from post-translational processing of the SLI-N309K protein. Western blots of the CRISPR-tagged parasites using the anti-*Pf*START1 antibody detected proteins of the expected size (Fig. 3D).

Upon addition of 2.5 mM glucosamine (GlcN) for 48 h *Pf*START1 was successfully knocked down in schizonts in both the SLI and the CRISPR parasites (Fig. 3B, E). The knockdown was stronger in the CRISPR constructs (75% for CR-WT, 81% for CR-N309K, 84% for CR-N330K) compared to the SLI parasites (60% for SLI-WT, 43% for SLI-N309K, 70% for SLI-N330K). This was expected as the *glm*S riboswitch is located further downstream of the *pfstart1* coding sequence in the SLI constructs compared to the CRISPR constructs.

To measure the impact of *Pf*START1 knockdown on parasite growth, ring-stage parasites (3D7, SLI- and CR- WT, N309K, N330K) were exposed to 0, 0.25, or 2.5 mM GlcN for 72 h, and their growth was assessed by LDH activity (Fig. S2C and F). In the presence of 2.5 mM GlcN, all SLI- and CR- parasites grew significantly less than in the absence of GlcN: approximately 40% less for most lines, and down to 50% less in the case of CR-N309K. 3D7 parasites were slightly impacted by the presence of 2.5 mM GlcN, but to a lesser extent, with ~5% reduced growth. The data therefore indicates that the reduction in the expression of *Pf*START1 reduces parasite growth.

To determine if the knockdown of *Pf*START1 also increased the sensitivity of the parasites to M-833, a growth inhibition assay was performed in the presence of 0, 0.25, or 2.5 mM GlcN, and the $EC_{50}$ calculated (Figs 3C, F). In both the SLI-WT and the CR-WT parasites, the

addition of 2.5 mM GlcN significantly reduced the $EC_{50}$ values: by 2.3-fold for more sensitive SLI-WT; 7.8-fold for CR-WT. In the case of the N309K mutation, the M-833 $EC_{50}$ values were reduced by similar amounts to the WT parasites in the presence of 2.5 mM GlcN (2.5-fold reduction for SLI-N309K, 5.2-fold reduction for CR-N309K), although this did not reach statistical significance. Regarding the N330K mutants, the addition of GlcN did not impact the $EC_{50}$ of M-833 in CR-N330K parasites. As expected, the addition of GlcN did not change the $EC_{50}$ of control 3D7 parasites. Together, these results confirm that M-833 targets *Pf*START1, and that *Pf*START1 is important for parasite growth.

### M-833-resistant parasites are also resistant to highly potent analogues

To investigate the therapeutic potential of the compounds, we explored the structure activity relationship of the M-833 series. It was observed that M-833 (**1**) shared structural similarities with the aryl aminoacetamide compound **2**[24] (Fig. 4A), which was developed from a Tres Cantos Antimalarial Set (TCAMS) hit. The structural similarities include a substituted aryl sulfonamide, acetamide core and aryl amido substitution in both series. As the mechanism of action for the Norcross et al. series has not been determined, its similarities to M-833 suggest it may share the same target. Compound **2** was assessed for activity against M-833-resistant populations and was shown to be cross-resistant (Fig. 4B), indicating compound **2** is likely to have the same molecular target as M-833. In the optimisation of M-833 it was found that there was a crossover with the structure-activity relationship between the two structurally related hit compounds[24]. Combining the gem dimethyl and the aryl oxy structural elements from M-833 and compound **2** respectively, and removal of the 2-methyl group from M-833 led to the intermediate hybrid compound **3** that exhibited a 10-fold improvement in antiparasitic activity ($EC_{50}$ = 120 nM) relative to the activity of M-833. Replacing the aryl oxy functionality in compound **3** with an amino group led to **W-991** (WEHI-991), resulting in a 20-fold improvement in parasite activity ($EC_{50}$ = 7 nM) (Fig. 4A). This change also led to a small increase in human HepG2 cell activity ($CC_{50}$ = 29 μM). This activity is consistent with that previously reported[24]. Methylation of the amido group resulted in an inactive analogue (compound **4**) ($EC_{50}$ > 10 μM) which was used as a negative control for our studies (Fig. 4A).

These newly synthesised analogues were tested against the original M-833 resistant parasites PopD-D7 and PopE-F10 which contain the *Pf*START1 N309K and N330K mutations respectively. Both parasite lines were highly resistant to the active analogues (compound **2**, **W-991**, and compound **3**) (Fig. 4B, Fig. S3 and Table S1). No change in $EC_{50}$ was observed between the resistant lines and 3D7 for the inactive analogue **4**.

Growth inhibition assays with GlcN were also carried out on the CRISPR parasites, using the potent analogue W-991 (Fig. 4C). The first observation was that the CR-N309K and CR-N330K parasites were around 600-fold less susceptible than the 3D7 and the CR-WT controls. This demonstrates that the *Pf*START1 mutations confer resistance to W-991. However, it should be noted that sub-micromolar potency was retained against the mutant parasites with an $EC_{50}$ below 700 nM in both cases. When *Pf*START1 was knocked down with the addition of GlcN, all CRISPR parasites became significantly sensitised to W-991 (7-fold for CR-WT, 5- fold for CR-N309K, 7.3-fold for CR-N330K), while the addition of GlcN did not change the $EC_{50}$ on control 3D7 parasites. These results show that we successfully developed potent analogues of M-833, that all appear to target *Pf*START1.

### M-833-series binds to recombinant *Pf*START1 protein and *Pf*START1 in parasites, but not to the mutant *Pf*START1(N330K) protein

To independently confirm that *Pf*START1 was the target for the M-833 series, we expressed recombinant wildtype *Pf*START1(WT) (I149-V394)

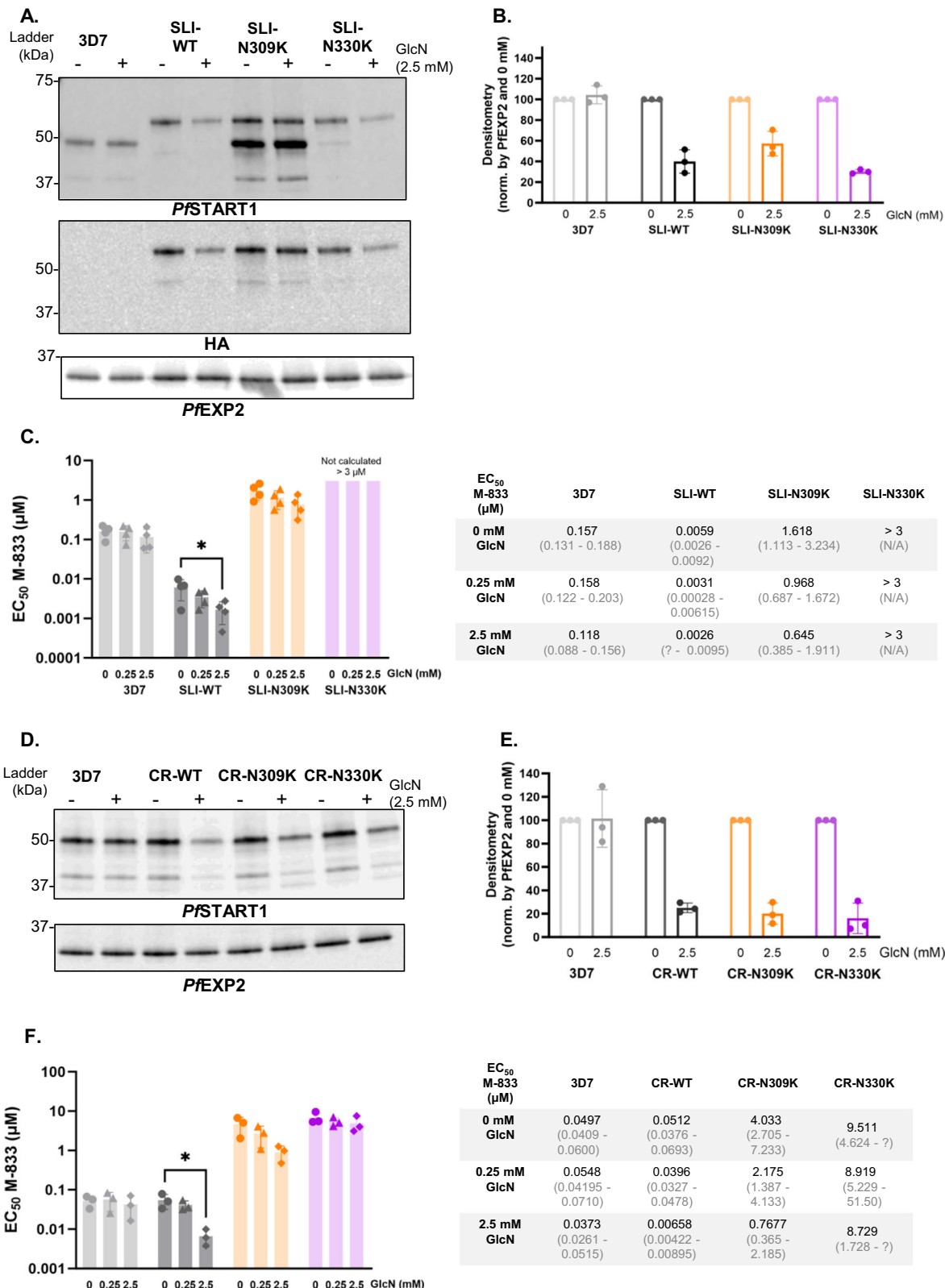

and measured binding to M-833, W-991 and negative control compound **4** by isothermal titration calorimetry (ITC) (Fig. 5A, Fig. S4A–C). M-833 bound *Pf*START1(WT) with nanomolar affinity ($K_D = 42 \pm 12$ nM), and was both enthalpically and entropically favoured, which is an optimal thermodynamic signature for inhibitor/target interactions (Fig. 5A). The optimised analogue, W-991, had improved binding affinity for *Pf*START1 ($K_D = 10 \pm 7$ nM) compared to M-833. This improved

binding affinity can be largely attributed to an enhanced ΔH, suggesting improved hydrogen bond contributions. *Pf*START1(WT) did not interact with the chemically related but inactive analogue **4**. To investigate the effect of the resistant mutations to compound binding we also recombinantly expressed *Pf*START1(N309K) and *Pf*START1(N330K) (I149-V394) (Fig. S4A, B). Since the former had a poor yield, only *Pf*START1(N330K) was evaluated in ITC binding

**Fig. 3 | Knocking-down *Pf*START1 sensitises parasites to M-833.** Engineered parasites contain a riboswitch *glm*S system: when glucosamine (GlcN) is added, *Pf*START1 is knocked-down. **A** 3D7, selection linked integrated (SLI)-wildtype (WT), -N309K or -N330K were exposed for 48 h to 0 or 2.5 mM GlcN, and proteins were extracted from schizonts. Western blots were probed with anti-*Pf*START1, anti-HA, and anti-*Pf*EXP2 antibodies. **B** The densitometry of 3 biological replicates was measured (the *Pf*START1 signal was normalised by the corresponding *Pf*EXP2 signal, and then normalised by the 0 mM GlcN condition within each parasite line; mean +/− SD). **C** $EC_{50}$ values of M-833 on 3D7 and SLI- parasites in the presence of 0, 0.25, or 2.5 mM glucosamine (GlcN). $n = 4$ biological replicates, mean +/− SD. Ordinary one-way ANOVA with Dunnett's multiple comparison test.

* adjusted $p$ value = 0.0334. $EC_{50}$ values and 95% confidence intervals are indicated in the table. **D** 3D7, CR-WT, -N309K, or -N330K were similarly exposed for 48 h to 0 or 2.5 mM GlcN, and proteins extracted from schizonts. Western blots were probed with anti-*Pf*START1 and anti-*Pf*EXP2 antibodies. **E** The densitometry of 3 biological replicates was measured (*Pf*START1 signal was normalised to the corresponding *Pf*EXP2 signal, and to the 0 mM GlcN condition; mean +/− SD). **F** $EC_{50}$ values of M-833 on 3D7 and CRISPR- parasites in the presence of 0, 0.25, or 2.5 mM glucosamine (GlcN). $n = 3$ biological replicates, mean +/− SD. Ordinary one-way ANOVA with Dunnett's multiple comparison test. *$p < 0.05$ ($p = 0.0137$). $EC_{50}$ values and 95% confidence intervals are indicated in the table. Source data are provided as a Source Data file.

experiments. This revealed no interaction with potent analogue W-991 (Fig. 5A, Fig. S4C), indicating the lysine substitute at the 330 position was the mechanism of parasite resistance in PopE parasites.

Next, to assess the specificity of the M-883 series we carried out drug-target engagement profiling in parasites. Here we leveraged the principle of Solvent Induced Protein Precipitation (SIP) which identifies protein-ligand interactions based on differential susceptibility of ligand-bound proteins to denaturation by organic solvents[25–27]. Here, soluble parasite protein lysate was exposed to increasing concentrations (0−25%) of a mixture of acetone, ethanol, and formic acid (AEF) in the presence of W-991 or DMSO, followed by soluble protein isolation via centrifugation. Soluble proteins were separated by SDS-PAGE and probed with an *Pf*START1 antibody (Fig. 5B). This showed that W-991 elicited protection to *Pf*START1 aggregation at 19-21% AEF (Fig. 5B, Fig. S4D). To further investigate the target engagement of W-991, we next turned to a global proteome analysis in the SIP assay. After the solvent challenge, soluble protein fractions were subsequently combined into two samples representing low and high denaturation pressure (7−15% and 17−25% AEF, respectively), following a Proteome Integral Solubility Alteration (PISA) experimental format[28]. Relative protein abundance was subsequently determined through Data Independent Acquisition mass spectrometry (DIA-MS) analysis (2,393 *Plasmodium* proteins with ≥3 peptides), followed by differential abundance analysis of drug- and vehicle-treated samples. This 'Solvent-PISA' assay revealed four proteins exhibiting significantly ($p < 0.01$) increased levels in the presence of W-991, of which *Pf*START1 was the most significant (Fig. 5C, $p = 0.0014$). The three other drug-stabilised proteins included merozoites-associated armadillo repeats protein (MAAP, PF3D7_1035900), ring-infected erythrocyte surface antigen (RESA3, PF3D7_1149200) and signal recognition particle subunit (SRP19, PF3D7_1216300) (Fig. 5C). The abundance/stability of the other two START-domain containing *P. falciparum* proteins detected in the assay was not affected by W-991 treatment (Fig. S4E). Taken together, the ITC and Solvent-PISA experiments strongly support *Pf*START1 as the principal molecular target of the M-833 series.

### *Pf*START1 inhibitors block ring development but their effect is reversible

M-833 treated parasites have been observed to invade normally but to stall before ring development[11]. To investigate whether parasites could recover from M-833 and W-991 treatment, parasites were treated similarly as in the Dans et al. study, and then followed-up for 3 days. Late-stage schizonts were treated for 4 h (during the egress/invasion window) with M-833, W-991, ML10 (reversible egress inhibitor), E64 (irreversible egress inhibitor), and heparin (invasion blocker), after which the compounds were removed, non-egressed schizonts eliminated with a sorbitol treatment, and parasite growth and phenotype monitored over 3 days (Fig. 6A, B). Growth was assessed every 24 h via bioluminescence of nanoluciferase (Nluc) in parasites expressing Hyp1-Nluc[29,30]. All treatments significantly reduced parasite growth compared to DMSO (Fig. 6B), with M-833, W-991, and heparin displaying the most striking difference, with virtually no recovery after three days (72 h). A short 4 h treatment was also sufficient to affect the

morphology of the treated parasites for days: at 72 h, M-833 and W-991-treated parasites resembled the dysmorphic ring-stage parasites described previously in the genetic knockout of *Pf*START1 (Fig. 6A)[11,17]. Based on the phenotype reminiscent of the *Pf*START1-knockout and the data shown in the paper, we are hereafter referring to M-833 and W-991 as *Pf*START1 inhibitors.

After confirmation that the M-833 series was blocking normal ring-stage development, we sought to determine (1) whether other intraerythrocytic stages were affected and (2) whether this resulted in parasite death or stasis beyond 72 h. To do this, we treated highly synchronous ring-stage parasites with DMSO, M-833 or W-991 and followed them through a cycle of growth, which did not indicate any impact on already-developed rings or trophozoites (Fig. S5A). We did, however, observe a slight developmental delay of schizonts upon compound treatment at 36 h. Despite this delay, the presence of dysmorphic rings in the M-833 and W-991-treatments was confirmed after 48 h (Figs. S5A, 6C). Drug-treated samples were then exposed to sorbitol lysis to remove unruptured schizonts and compounds were washed out. Parasites were then returned to culture alongside their continuously treated counterparts. This revealed that after 6 days post washout ($T = 192$ h), parasites that had been pre-treated with either M-833 or W-991 were able to resume normal growth, which was visible by both the increase in parasitemia as measured by SYBR Green staining (Fig. 6D) and Giemsa-stained blood smears (Fig. 6C), indicating that one cycle of treatment is not sufficient to cause irreversible inhibition of parasite growth. To corroborate this, we performed parasite reduction ratio (PRR) assays whereby the parasites were exposed to W-991 for up to 5 days before the compound was removed and parasites returned to culture for 21 days. This revealed that despite a reduction of viable parasites after an increasing exposure time to W-991, a proportion of parasites remained after a treatment period of 5 days (Fig. S5B). Taken together, these experiments show that complete inhibition of in vitro parasite growth with the *Pf*START1 inhibitors requires >5 days of continuous treatment, and parasite survival with <5 days of treatment is likely due to stasis in the ring-stage of the asexual lifecycle.

### *Pf*START1 inhibitors block differentiation into ring-stage parasites by preventing PVM expansion

It has been previously postulated that the phospholipid transfer activity of *Pf*START1 may be required for the expansion of the parasitophorous vacuole (PV) after invasion[16]. Since previous live cell imaging of merozoites treated with M-833 also showed a defect in ring-stage establishment[11], we utilised lattice light sheet microscopy to visualise PVM formation directly after merozoite invasion in the presence of our potent inhibitor W-991. To track these events, the RBC membrane was labelled with a fluorescent steryl dye, Di-4-ANEPPDHQ, and merozoites with Mitotracker Deep Red dye[31]. Consistent with previous reports[9,31], several minutes after invasion in the DMSO control, the PVM was observed to become less spherical and more irregularly shaped consistent with the differentiation of the spherical merozoite into an amoeboid ring (Fig. 7A, Movie S1). In contrast, newly invaded merozoites treated with W-991 remained spherical over this period (Fig. 7A, Movie S2). Bioinformatic analysis of 4-dimensional

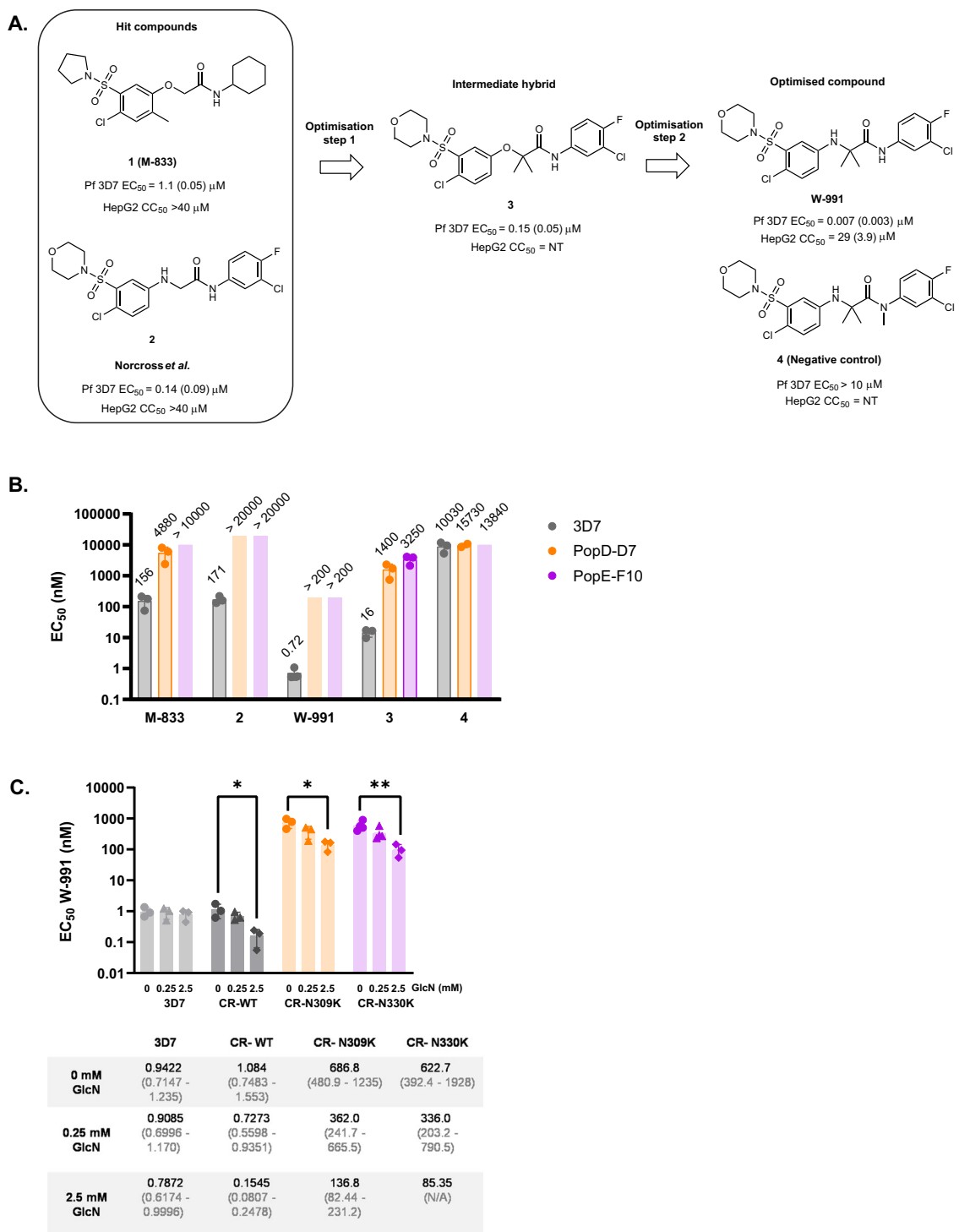

**Fig. 4 | Analogues of M-833 are resisted by M-833 resistant lines, SLI- and CRISPR-mutants. A** Optimised analogues of M-833: their structures, $EC_{50}$ (SD) on 3D7 parasites, toxicity ($CC_{50}$ (SD)) on HepG2 cells. NT = not tested. **(B)** $EC_{50}$ values of M-833 and analogues on 3D7 and M-833 resistant parasites PopD-D7 and PopE-F10, containing the PfSTART1 mutations N309K and N330K, respectively. $n = 3$ biological replicates, mean +/− SD. $EC_{50}$ values indicated above the bar, as well as in Supplementary Data 1. Growth curves available in Fig. S3. **C** $EC_{50}$ values of W-991 on

3D7, CR-WT, CR-N309K and CR-N330K parasites, in the presence of 0, 0.25 or 2.5 mM glucosamine (GlcN). $n = 3$ biological replicates, mean +/− SD. Ordinary one-way ANOVA with Dunnett's multiple comparison test. *$p < 0.05$. **$p < 0.005$ (CR-WT, $p = 0.0265$; CR-N309K, $p = 0.0123$; CR-N330K, $p = 0.008$). $EC_{50}$ values and 95% confidence intervals are indicated in the table. Source data are provided as a Source Data file.

images of nascent PVMs indicated the mean sphericity of the PVM of W-991-treated merozoites stayed high during the observation period compared to the DMSO control where the PVM became less spherical consistent with change in shape to accommodate the amoeboid ring (Fig. 7B, Fig. S6A, $p = 0.045$ at 15 min). The mean vacuole surface area

and volume of the PVM were also reduced in W-991 but this did not reach significance (Fig. S6B, C; $p = 0.109$, $p = 0.129$, respectively). Overall quantification of the lattice light sheet microscopy supports our earlier empirical observations that inhibition of *Pf*START1 prevents normal development of the parasite plasma membrane and/or PVM to

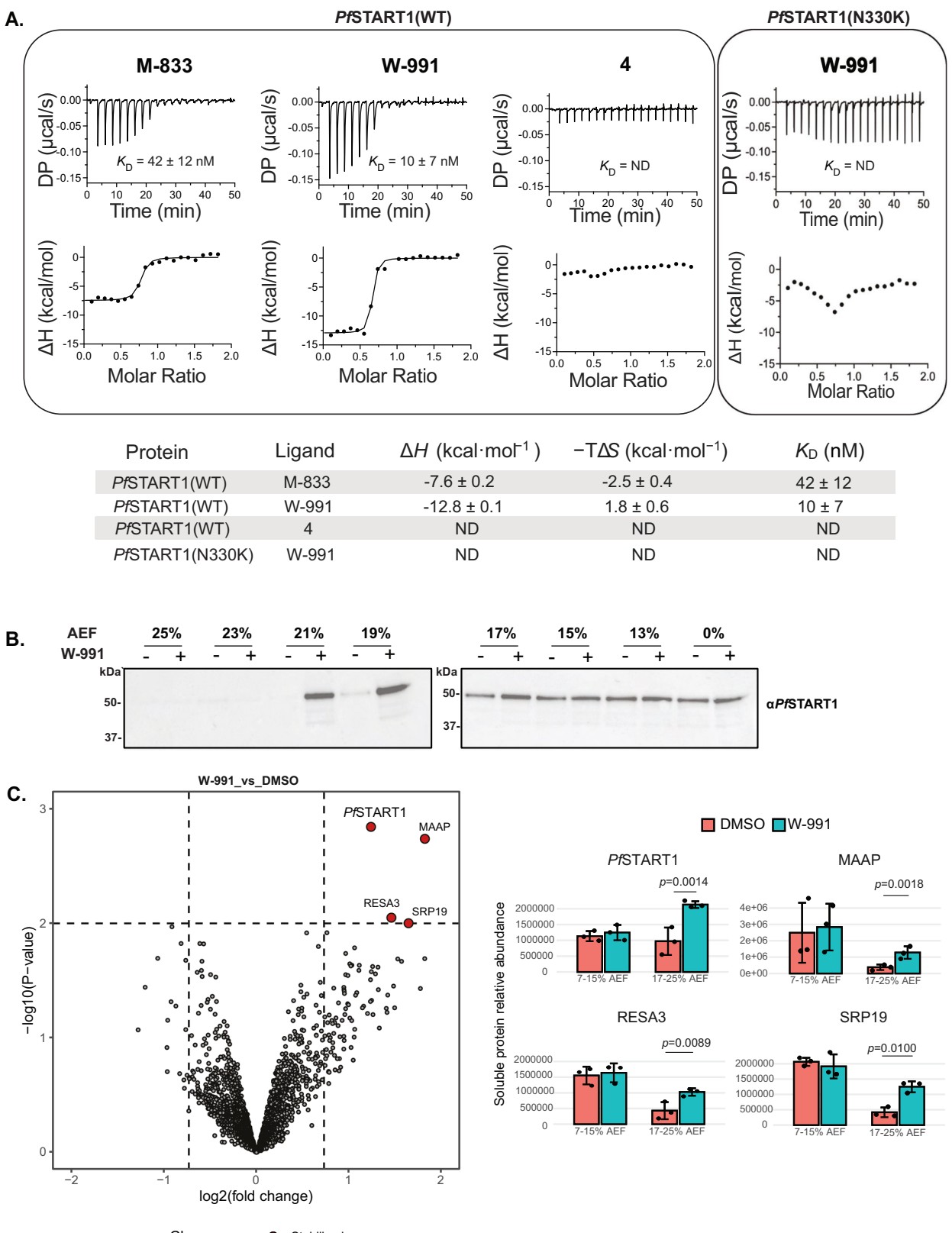

| Protein | Ligand | $\Delta H$ (kcal·mol$^{-1}$) | $-T\Delta S$ (kcal·mol$^{-1}$) | $K_D$ (nM) |
|---|---|---|---|---|
| *Pf*START1(WT) | M-833 | -7.6 ± 0.2 | -2.5 ± 0.4 | 42 ± 12 |
| *Pf*START1(WT) | W-991 | -12.8 ± 0.1 | 1.8 ± 0.6 | 10 ± 7 |
| *Pf*START1(WT) | 4 | ND | ND | ND |
| *Pf*START1(N330K) | W-991 | ND | ND | ND |

help expand the PV space, thereby reducing the capacity of the merozoite to differentiate into a ring-stage parasite.

### *Pf*START1 inhibitors do not prevent sporozoite invasion but block parasite transmission to mosquitoes

Since sporozoites are known to form a PV in hepatocytes after invasion[32], we evaluated W-991 against *P. berghei* sporozoite invasion

to determine if *Pf*START1 ortholog may be required for this process. This revealed that W-991 had no impact against *P. berghei* sporozoite invasion (Fig. S7A, *p* > 0.05).

Next, we assessed the activity of the M-833 series against the sexual stage of the *P. falciparum* lifecycle, in a *P. falciparum* dual gamete formation assay (DGFA)[33]. Mature gametocytes were treated for 48 h with 1 µM M-833 or W-991 and then gametogenesis triggered.

**Fig. 5 | The M-833 series demonstrates target engagement for *Pf*START1 while the mutant form prevents compound binding. A** Isothermal titration calorimetry (ITC) analysis of M-833 series and recombinant *Pf*START(WT) or *Pf*START(N330K). Representative thermograms of 90 μM *Pf*START1 titrated into 10 μM M-833, W-991, and negative control **4**. The bottom panel comprises the data after integration of the peaks and a fitted offset applied. The binding curve shows the fit to a single-site binding model. DP = differential power. The table represents summary of the mean affinity, enthalpy, and entropy obtained for M-833, W-991 and **4** binding to *Pf*START1 from *n* = 2 experiments. Error represents standard deviation. ND = not determined. The second replicate and values for individual parameters are provided in Fig. S4C. **B** Solvent proteome profiling assays. Parasite lysate was treated with DMSO or 10 μM W-991, challenged with an acetic acid/ethanol/formic acid mixture (AEF) and soluble fractions extracted and analysed via western

blots probed with anti-*Pf*START1. Western blot replicates are shown in Fig. S4D. **C** Volcano plots depict differential soluble protein abundance analysis (moderated t-test based on limma package) of parasite lysate treated with W-991 (100 μM) or the DMSO control (*n* = 3 biological replicates, mean +/− SD) after solvent-induced protein precipitation (7–25% AEF). Non-significant (ns) proteins are plotted in grey, and significant stabilised proteins are in red. Hit selection cut-offs of 0.73 log2 fold-change and *p* < 0.01 are indicated with dashed lines. Significant stabilisation hits are shown in the bar graphs representing the average of three biological replicates of relative soluble protein abundance in DMSO- or W-991-treated parasite lysate samples after solvent-induced protein precipitation, plotted for Gradient 1 'G1' (7–15% AEF) and Gradient 2 'G2' (17-25% AEF). Error bars represent standard deviation. Source data are provided as a Source Data file.

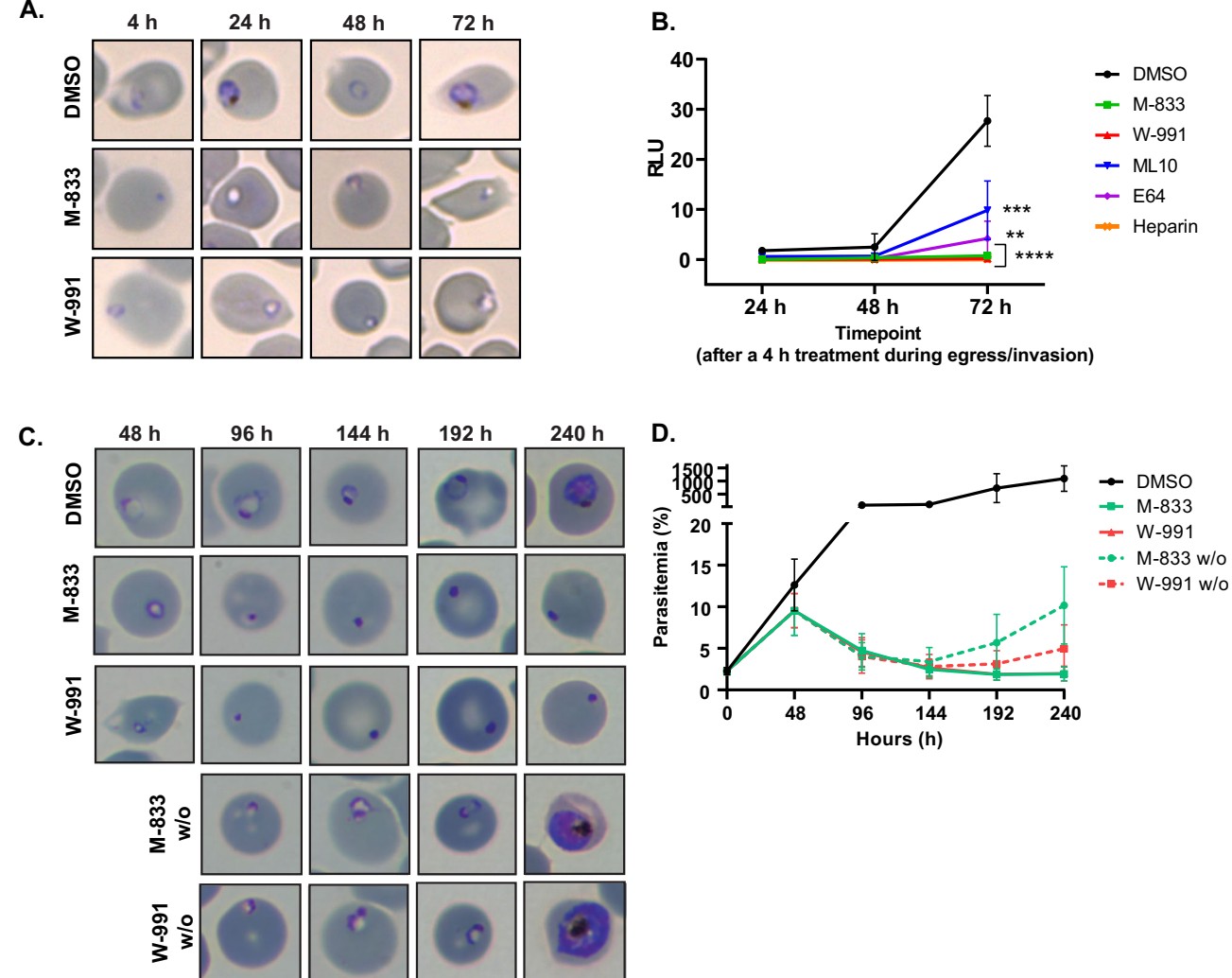

**Fig. 6 | *Pf*START1 inhibitors block ring development but their effect is reversible. A**, **B** Tightly synchronised Hyp1-Nluc schizonts were exposed to the following drugs for 4 h (during the egress/invasion window): M-833 (2 μM), W-991 (60 nM), DMSO (0.02%), ML10 (30 nM) (egress inhibitor), E64 (10 μM) (irreversible egress inhibitor), and heparin (100 μg/mL) (invasion inhibitor). After 4 h, non-egressed schizonts were eliminated with a sorbitol treatment, the drugs washed off, and parasites were followed over 72 h. Smears were taken at each timepoint (**A**) and growth was assessed with a bioluminescence read-out in relative light units (RLU x 10⁵) (**B**). *n* = 3 biological replicates, mean +/− SD. Ordinary one-way ANOVA with Tukey's multiple comparison test was conducted on the 72 h timepoint. *\*p* < 0.05

(ML10 vs. Heparin; not shown on the graph, *p* = 0.0484). \*\*\**p* < 0.005 (DMSO vs. ML10, *p* = 0.0006; DMSO vs. E64, *p* = 0.0002). \*\*\*\**p* < 0.0001 (DMSO vs. M-833, W-991 and heparin). Highly synchronous ring-stage 3D7 parasites were exposed to M-833 (2 μM), W-991 (60 nM) or DMSO (0.02%). After 48 h, populations of M-833 and W-991-treated parasites were treated with sorbitol, compounds were washed out (w/o) (dotted lines), and morphology visualised by Giemsa-stained thin blood smears (**C**) and parasitemia was quantified by SYBR Green staining and flow cytometry (**D**) every 48 h for four following cycles of growth. Error bars indicate the standard deviation of two biological replicates, each made up of three technical replicates. Source data are provided as a Source Data file.

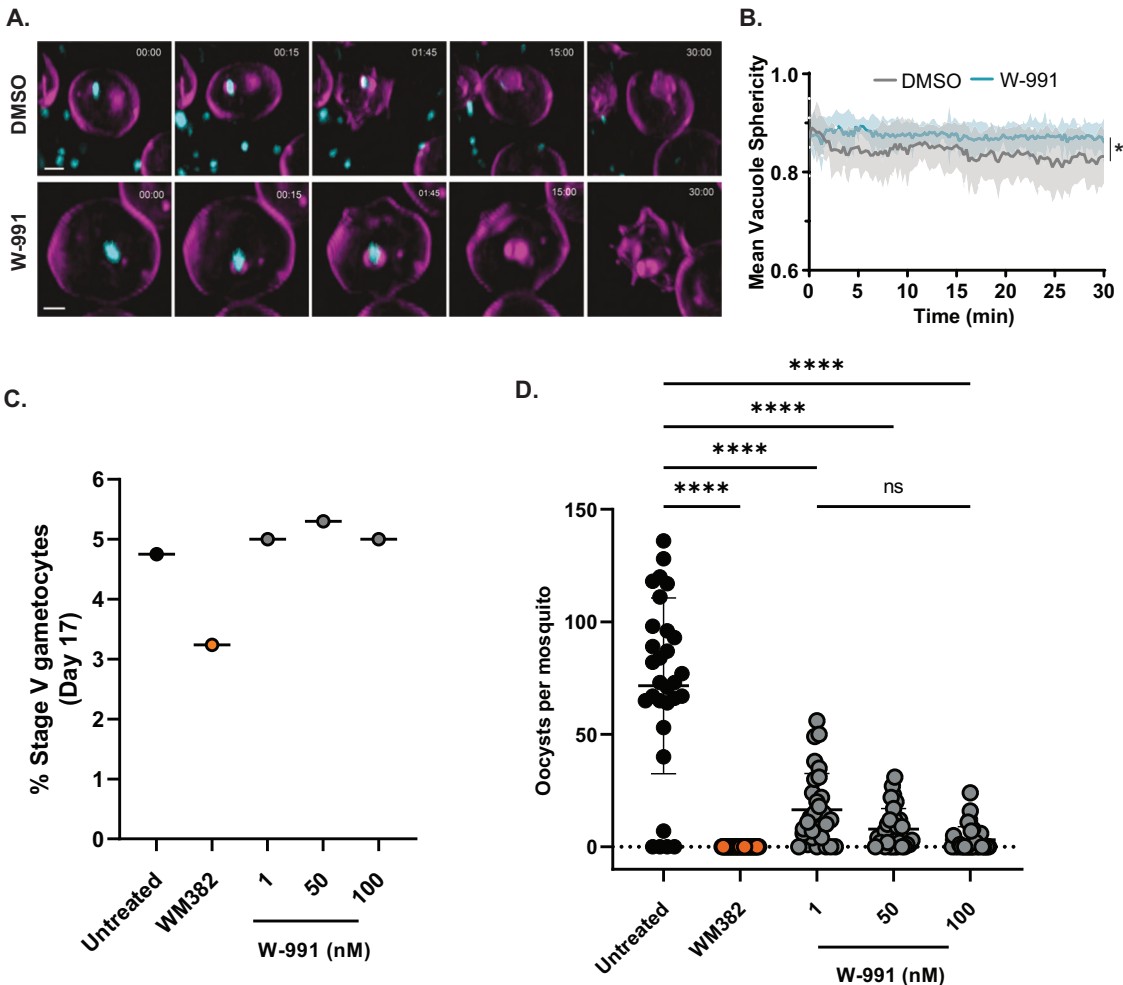

**Fig. 7 | W-991 blocks merozoite transition to ameboid ring in RBCs, and blocks transmission in mosquitoes. A** Representative images of 4D Lattice Light Sheet Microscopy of merozoites (MitoTracker; cyan) directly after invasion of RBCs (all membranes are magenta). Treatment with W-991 shows disruption to the formation of the parasitophorous vacuole membrane (magenta) compared to vehicle control. **B** The mean vacuole sphericity (+/− SD) for W-991 and DMSO treatments across 30 min filming period. Statistical analysis was conducted via a Nested t-test (two-tailed) in GraphPad Prism showing $p = 0.716$ at $T = 0$ mins and a $p = 0.045$ at $T = 15$ mins (*). Across three independent experiments, there were 20 and 22 events analysed for W-991 (60 nM) and DMSO (0.001%) treatments, respectively (Fig. S6 and Movie S1 and S2). **C** Stage V gametocytes were quantified on day 17 which demonstrates there was no defect in gametocyte development upon W-991 treatment compared to PMIX/X dual inhibitor WM382. **D** Oocysts were quantified in mosquito midguts on day 7 after feeding which showed a dose-dependent decrease in oocysts per mosquito with W-991 treatment. Statistical tests (two-way ANOVA) were performed in GraphPad Prism. **** indicates $p < 0.0001$. Positive control compound, WM832, was used at 50 nM[53]. Error bars indicate SD for n = 30 mosquitoes for each treatment (mean+/−SD). Data are a single representative from 3 independent experiments which can be found in Figure S7C, D. Source data are provided as a Source Data file.

Exflagellation and surface Pfs25 expression were quantified, which demonstrated that no inhibition of either male or female gamete formation in the presence of M-833 and W-991 had occurred (Fig. S7B).

To further evaluate this series in the sexual stage we next performed a standard membrane feeding assay (SMFA) where 3 increasing concentrations of W-991 (1, 50, 100 nM) were exposed to stage IV-V gametocytes from days 14-17. On day 17 the percentage of stage V gametocytes were evaluated which showed there was no impact of W-991 affecting gametocyte development (Fig. 7C). Media containing the compounds was then removed and treated gametocytes were fed to mosquitoes and the number of oocysts present in the midgut of mosquitoes was quantified 7 days later. This showed that W-991 exhibited transmission blocking activity of *P. falciparum* to mosquitoes, with 100 nM of W-991 reducing the number of oocysts by >20-fold compared to untreated control (Fig. 7D, $p > 0.001$). To corroborate this result, we repeated the SMFA using another insectary facility which also demonstrated a reduction in transmission upon the same doses of W-991 (Fig. S7C-D), indicating that the *Pf*START1 inhibitors

cause irreversible inhibition of an essential process between induction of gametocytogenesis and oocyst formation, thereby showing potential as a transmission blocking antimalarial.

## *Pf*START1 expression, processing, and localisation in schizonts and merozoites

To better understand the biological role of *Pf*START1, SLI-WT parasites were tightly synchronised and western blots were performed on various parasites stages (Fig. 8A, Fig. S8A). Following normalisation to the *Pf*HSP70-1 loading control, *Pf*START1-HA was most strongly expressed in schizonts and in very young rings as observed previously[16] (Fig. 8B). In addition to the main ~60 kDa band (corresponding to PEXEL-cleaved *Pf*START1-HA), a minor processed band (around 50 kDa) was also observed (Fig. 8A). The site of cleavage is not known, but must occur near the N-terminal, since this processed band was also detected with the HA-antibody.

To further understand the processing of *Pf*START1, proteins of tightly synchronised wildtype 3D7 schizonts without the HA tag were

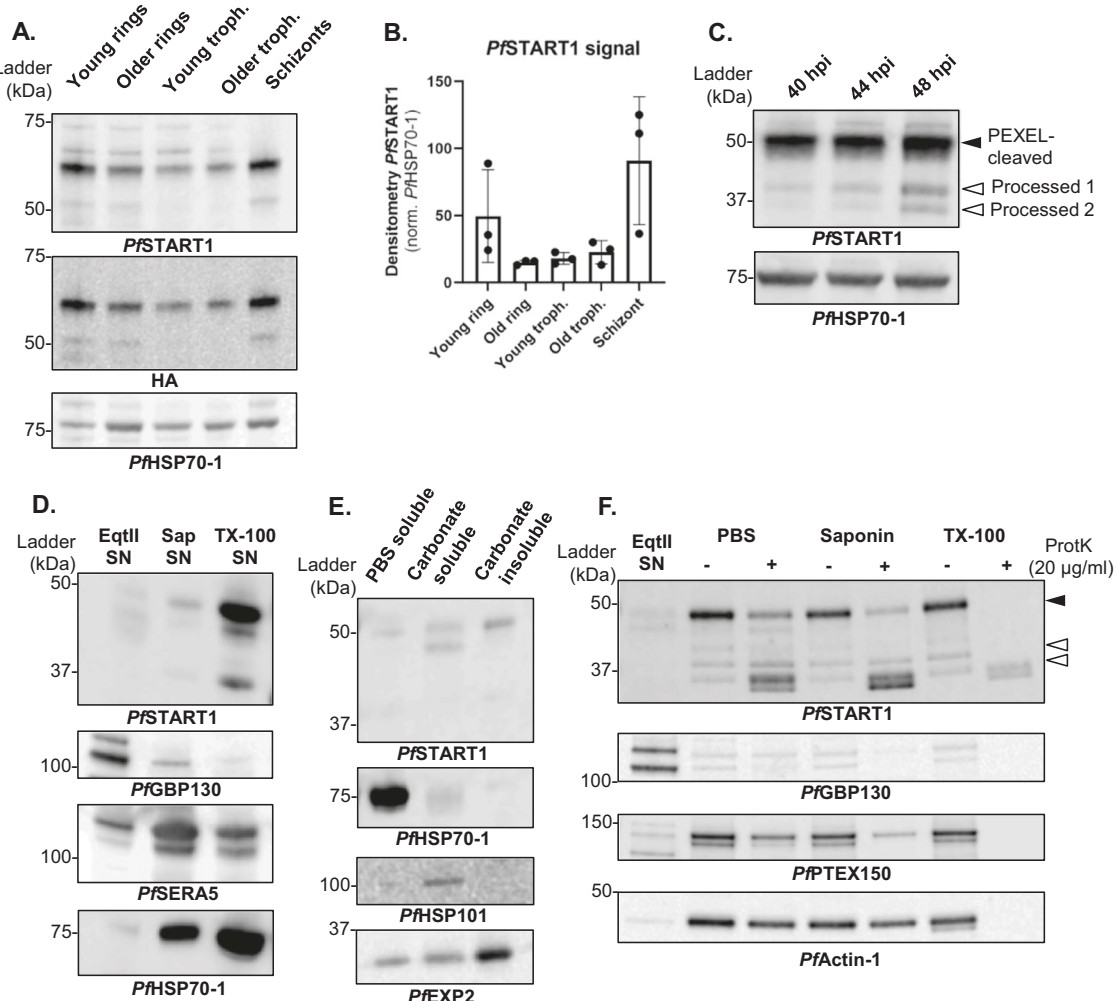

**Fig. 8 | *Pf*START1 expression and processing. A** Western blot of SLI-WT parasites along the 48 h erythrocytic cycle. A saponin-lysis was performed on tightly synchronised SLI-WT young rings, older rings, young trophozoites, older trophozoites, and schizonts. **B** Densitometry of *Pf*START1 over three biological replicates (normalised by the corresponding *Pf*HSP70-1 signal; mean +/− SD). **C** Synchronous 3D7 trophozoites were magnet-purified, and whole cells harvested at ~40 hpi (hour post invasion), ~44 hpi and ~48 hpi (this last sample was treated with 10 μM E64 to prevent egress). PEXEL-cleaved *Pf*START1 and two further processed forms are indicated by solid and empty arrows respectively. **D** To localise *Pf*START1, Percoll-purified 3D7 schizonts were sequentially lysed with equinatoxin II (EqtII), saponin (Sap) and Triton-X100 (TX-100), and the supernatants (SN) were collected to harvest proteins localising in the red blood cell (RBC) cytosol, the parasitophorous vacuole (PV) and the parasite, respectively. *Pf*GBP130 is a protein known to be exported into the RBC cytosol; *Pf*SERA5 localises to the PV in schizonts; *Pf*HSP70-1 is a parasite cytosolic protein. **E** To determine the solubility of *Pf*START1, saponin-lysed 3D7 schizonts were sequentially lysed in PBS (with five freeze-thaw cycles) and sodium carbonate. The supernatant resulting from the PBS lysis was collected (PBS soluble), as well as the supernatant and the pellet of the carbonate lysis (Carbonate soluble and Carbonate insoluble respectively). Controls are the soluble protein *Pf*HSP70-1, the membrane-associated protein *Pf*HSP101, and the integral transmembrane protein *Pf*EXP2. **F** Proteinase K protection assay was conducted on Percoll-purified 3D7 schizonts. Schizonts were first lysed in EqtII, the supernatant (SN) of which was collected. The remaining parasite and PV were either incubated in PBS (no lysis), saponin (PVM-lysis), or TX-100 (lysis of all membrane) with or without proteinase K. *Pf*GBP130 is RBC cytosolic protein; *Pf*PTEX150 is a PV protein; *Pf*Actin-1 is a parasite cytosolic protein. Note that another replicate for each of these experiments is shown in Fig S8. Source data are provided as a Source Data file.

analysed every 4 h (E64 was used for the latest timepoint to prevent parasite egress; Fig. 8C). This time course demonstrates that while the majority of *Pf*START1 remains as a ~48 kDa band (PEXEL-cleaved), two further processing events occur as schizogony progresses (*Pf*START1$_{proc1}$~39 kDa, *Pf*START1$_{proc2}$ ~ 36 kDa). The processing of *Pf*START1-HA occurred at the N-terminus, as both processed bands could be detected by an HA-antibody on SLI-WT schizonts (Fig. S8D). To more precisely pinpoint when *Pf*START1-HA was processed during schizogony, we treated schizonts with inhibitors stalling egress at different time points (Fig. S8E-F): 49c is a plasmepsin X (PMX) inhibitor, which prevents activation of SUB1, early in the egress pathway[34]; compound 1 (C1) is a protein kinase G inhibitor, which prevents exoneme discharge[35]; E64 is a cysteine protease inhibitor which prevents the rupture of the red blood cell membrane, near the very end of

egress[36]. The densitometry of processed bands was normalised to the PEXEL-cleaved *Pf*START1, and the effect of treatment was analysed (Fig. S8F). 49c-treated schizonts contained the least amount of *Pf*START1$_{proc1}$ (compared to DMSO and other treatments), and C1-treated schizonts had the lowest levels of *Pf*START1$_{proc2}$. E64-treated schizonts contained most of both processed bands, with a significant enrichment in *Pf*START1$_{proc2}$ compared to the DMSO control.

*Pf*START1 contains an unusual PEXEL motif and was recently described as not being exported to the RBC in trophozoites[37]. To investigate *Pf*START1 localisation in schizonts, 3D7 schizonts were Percoll-purified and sequentially lysed in equinatoxin II (EqtII), saponin (Sap) and Triton X100 (TX100) to collect the supernatant (SN) corresponding to the RBC cytosol soluble fraction, the parasitophorous vacuole (PV) soluble fraction, and the parasite fraction, respectively

(Fig. 8D and Fig. S8G). *Pf*GBP130, a known exported soluble protein[38,39], was detected in the EqtII SN; *Pf*SERA5, a soluble protein that localises in the PV of schizonts[38,40], was enriched in the Sap SN; parasite cytosolic protein *Pf*HSP70-1 was found mainly in the TX-100 SN. *Pf*START1 was most strongly detected in the TX100 fraction, indicating that it was either within the parasite, and/or associated with membranes.

We tested whether *Pf*START1 was fully soluble (in which case, *Pf*START1 should be localised within the parasite), membrane-associated or membrane-bound. To do so, saponin-lysed 3D7 schizonts were sequentially lysed: first in PBS (with freeze-thaw cycles to break the cells), the supernatant of which was described as PBS soluble. The pellet was then incubated in sodium carbonate: the supernatant was described as carbonate soluble (proteins associated with membranes) and the pellet as carbonate insoluble (integral membrane proteins) (Fig. 8E and Fig. S8H). Controls included *Pf*HSP70-1, *Pf*HSP101, and *Pf*EXP2, which were detected in the PBS soluble, the carbonate soluble, and the carbonate insoluble fractions, respectively. *Pf*START1 was mainly detected in the carbonate soluble and insoluble fractions in accordance with previous findings[16].

We then performed a proteinase K protection assay to indicate where *Pf*START1 was localised after lysing different membranes as per Fig. 8E, −/+ proteinase K (Fig. 8F, Fig. S8I). *Pf*GBP130 was localised in the RBC as it was detected in the EqtII supernatant. Cytoplasmic *Pf*Actin-1 was mostly protected from proteinase K degradation unless all the membranes were lysed in TX-100. *Pf*PTEX150 is a PV protein[41], and as such we expected it to be degraded mainly upon the addition of proteinase K and saponin. In our assay however, *Pf*PTEX150 was slightly degraded by proteinase K in PBS, further degraded in saponin and completely degraded in TX-100. This suggests that the PVM in our schizonts might have been partially compromised in the PBS treatment. The incomplete degradation of *Pf*PTEX150 in the saponin condition suggests either that the saponin lysis was not complete and some PVM remained unruptured, or that a fraction of PfPTEX150 remains inside the parasite (newly synthesised, not yet secreted into the PV). A pattern identical to *Pf*PTEX150 was observed for PEXEL-cleaved *Pf*START1, suggesting that more of this form of the protein is in the PV than in the parasite. *Pf*START1$_{proc1}$ appears to reside within the parasite as it seems to be protected from proteinase K cleavage unless TX-100 was added. For *Pf*START1$_{proc2}$, a similar conclusion is harder to draw considering the degradation products co-migrate in the same size. Altogether this data corroborates previous work[12,16,17] that indicates *Pf*START1 is most strongly expressed in schizonts, associated with membranes, and that more is located in the PV than in the parasite.

## Discussion

In this study, we explored the activity of the anti-*Plasmodium* M-833 compound series where whole genome sequencing of resistant parasites revealed point mutations in *Pf*START1 (N309K, N330K, I224F). Through structure-activity relationship studies, we produced analogues that remained on-target for *Pf*START1, whilst significantly increasing the potency to achieve an EC$_{50}$ against wildtype parasites in the low nanomolar range. Engineering mutations N309K and N330K into drug sensitive parasites conferred resistance to M-833 and to the potent analogue W-991. Furthermore, knocking down *Pf*START1 sensitised parasites to both M-833 and W-991. We demonstrated the direct binding of M-833 and analogues to *Pf*START1, using both recombinant and native proteins. Overall, this demonstrates that M-833 and analogues exert their antiparasitic activity by targeting *Pf*START1.The data presented here strongly suggest that the compounds studied inhibit *Pf*START1's lipid transfer activity, but it will be important in the future to confirm this using specific transferase activity assays.

Mammalian START proteins are known to have pathological roles in several human diseases (e.g., STARD3 overexpression in cancer[42]),

including infectious diseases (e.g., human STARD11 is hijacked by intracellular *Chlamydia trachomatis* bacteria[43]). Due to this, several inhibitors of human START proteins have been identified: STARD1 inhibitors C1-C6[44]; STARD11 inhibitors HPA-12[45] and SC1 plus analogues[46]; STARD3 (predicted) inhibitor D(-)-tartaric acid[47] and STARD3 inhibitor VS1 (low potency)[48]. The M-833 series does not show structural similarity to the aforementioned human START inhibitors and considering that the M-833 series had CC$_{50}$ > 29 μM against HepG2 cells (a cell line known to express START proteins[49]), this indicates that our compounds are unlikely to target human START proteins. Additionally, Solvent PISA profiling suggests target specificity of M-833 series for *Pf*START1 over other *P. falciparum* START proteins (PF3D7_1351000 and PF3D7_1463500) detected in the assay. Of note, the other proteins stabilised in the Solvent PISA assay (*Pf*MAAP, *Pf*RESA3 and *Pf*SRP19) did not contain any mutations in the initial resistance selection suggesting protein binding to W-991 occurs without contributing to the mode of action of the compound.

Solvent PISA profiling described here represents a novel unbiased strategy for antimalarial drug-target identification. In addition to representing a means of orthogonal protein-engagement validation for target-candidates derived from parallel approaches, such as thermal proteome profiling, it enables the detection of drug interactions with proteins not susceptible to temperature-induced aggregation. *Pf*START1 represents one of such highly thermostable protein[50], and its interaction with the drug in parasite lysate cannot be identified based on stability shift <75 °C. Further, the Solvent PISA strategy introduced here involves major improvements over mass spectrometry-based target identification workflows established for *Plasmodium*[50-53]. Incorporation of Proteome Integral Solubility Alteration (PISA) assay format[54] and Data Independent Acquisition mass spectrometry (DIA-MS)[55] into the solvent assays significantly reduces the required MS-analysis time and profiling cost, as well reducing sampling bias by profiling the entire proteome across the solvent gradient.

It was initially encouraging to observe that a short 4 h treatment stopped parasite development for three days post compound removal. However, a longer follow-up of M-833 and W-991 treated (and removed) parasites indicated that they could recover over a longer period, which was confirmed by the parasite reduction ratio assay. Due to the essentiality of *Pf*START1 in the blood stage[17], it was somewhat unexpected that parasites could recover after being subjected to 10 x EC$_{50}$ of W-991 for up to five days. One possible explanation of this is that the EC$_{50}$ value derived from a standard 72 h growth assay could underestimate the concentration required to effectively kill parasites due it being unable to differentiate between stalled merozoite conversion into rings versus dead parasites. It is therefore plausible that conducting a longer growth assay (>72 h) would assist in determining a dose that would cause irreversible death. Alternatively, it is also possible that inhibiting *Pf*START1 does not lead to death but rather the parasites entering a dormant mode in which they can survive until compound levels have decreased. This is reminiscent of artemisinin-resistant parasites whereby prolonged ring-stage has been associated with treatment survival[56,57]. Interestingly, *Pf*START1 has been recently found to be significantly upregulated in both lab-adapted and field-derived artemisinin-resistant strains that contain mutations in Kelch13 (Dd2$^{C580Y}$ and Cam3.II$^{R539T}$)[58]. Additionally, artemisinin resistance has also recently been shown to be present in male gametocyte activation whereby parasites resistant to artemisinin are activated under artemisinin treatment, whilst sensitive parasites remain inactivated[59]. It would, therefore, be interesting to conduct combination experiments with our *Pf*START1 inhibitors against artemisinin-resistant strains in both the asexual blood stage and mosquito stages to determine if inhibiting *Pf*START1 could sensitise resistant parasites to artemisinin.

The activity profile of the *Pf*START1 inhibitors was found to be variable in other stages of the lifecycle outside the blood stage with no drug effect against liver stage invasion, stage V gametocyte

development, and gametogenesis. The lack of activity against the liver stage of infection was surprising given that sporozoites grow and replicate in a PV, similar to the asexual blood stages[60,61]. The apparent inactivity of the PfSTART1 inhibitors could be due to the experimental design where *P. berghei* sporozoites and W-991 were added to human liver cells and evaluated only 2 h post invasion. It is possible we did not capture the inhibition of PV formation and further experiments where sporozoites are added to liver cells with the compounds for a longer period may indicate growth inhibition.

We found that PfSTART1 inhibitors were active in a standard membrane-feeding assay. Whilst the functionality of PfSTART1 has been previously explored in the asexual blood stage[16,17,37], there is little known about PfSTART1 in the mosquito stage of infection. Transcriptomic studies have shown that PfSTART1 is expressed in salivary gland sporozoites[62,63] and ookinetes[64]. Since there is no PV formation in ookinetes[60,65,66], it is possible that PfSTART1 is utilised for another lipid-dependent process during this stage. Sporozoites, however, are known to form a PV during the invagination of the epithelial cell membrane in the mosquito salivary glands[67–69] which could suggest that the PfSTART1 inhibitors could act through a similar mode of action to that we see in the asexual blood stage but further investigations are required to confirm this.

To study the function of PfSTART1 in the blood stage, we tagged the C-terminal region of the protein with a 3 x HA tag using the selection-linked integration (SLI) system. Unexpectedly, we saw an increased sensitivity of the SLI-WT parasites which did not occur in the CRISPR WT parasites that did not include a tag. This may be due to the addition of the 3 x HA and P2A peptide (additional 55 amino acids) to PfSTART1, which could be interfering with its normal functioning and thereby sensitising the SLI-WT parasites to M-833. The C-terminus of PfSTART1 has been shown to be important for the regulation of lipid transfer[17], with the last 30 amino acids of PfSTART1 predicted to form an alpha helix (Alphafold)[70,71].

Consistent with previous reports, we found that PfSTART1 is predominantly expressed in schizonts in the asexual blood stage, is membrane-associated and is not exported into the surrounding RBC (despite its cleaved PEXEL motif)[16,37]. The close association with membranes is probably mediated by one or more interacting partner(s), considering that PfSTART1 does not contain a transmembrane domain. PfSTART1 localises partly to the PV and partly within the parasites in schizonts which suggests that PfSTART1 may have several functions, transferring lipids in several locations.

PfSTART1 appears to be processed multiple times at its N-terminus. Fréville and colleagues demonstrated that the PEXEL motif of PfSTART1 is cleaved by plasmepsin V, but that PfSTART1 is not exported[37]. Therefore, the main PfSTART1 signal observed on western blots at ~48 kDa likely corresponds to the PEXEL-cleaved protein. In addition, we demonstrated that two further processing events occur as schizogony progresses. First, the processing to PfSTART1$_{proc1}$ (~39 kDa) seems to occur early in schizogony, shortly after SUB1 activation by plasmepsin X, because plasmepsin X inhibition reduced the levels of PfSTART1$_{proc1}$. Interestingly, PfSTART1 contains a plasmepsin X cleavage site (SDIQ[72]), although the resulting theoretical products should be 20 kDa and 26 kDa whilst bands at 39 kDa and 36 kDa were observed. No SUB1 or plasmepsin IX cleavage sites were identified in PfSTART1[53,73]. The second processing event to PfSTART$_{proc2}$ (~36 kDa) appears to occur after protein kinase G activation and accumulates considerably in the presence of cysteine inhibitor E64. Overall, it is possible that these processed forms of PfSTART1 are important in merozoites for the upcoming invasion and establishment of the PVM. PfSTART1$_{proc1}$ (and possibly PfSTART1$_{proc2}$) show some resistance to proteinase K after saponin lysis, indicating they are not located within the PV like the PEXEL-cleaved protein. However, both PfSTART1$_{proc1}$ and PfSTART1$_{proc2}$ only represent a small proportion of the total PfSTART1 (less than 40% of the PEXEL-cleaved form). Therefore, the functional relevance of these processed PfSTART1 remains to be investigated.

Inhibition of PfSTART1 in the blood stage led to the abnormal development of the membranes surrounding the merozoite directly after invasion as visualised with lattice light sheet microscopy. Whilst this phenotype was distinct, it remains unknown as to the composition of these lipids, in addition to the direction and location of transport. In *Plasmodium*, the molecular mechanisms underpinning PVM formation are still unknown[74,75] but it has recently been shown that the PVM is comprised of mostly host RBC lipids, rather than parasite-derived material[31]. Overall, our data supports a model that has been previously proposed[17] whereby PfSTART1 aids in expanding the PVM using lipids from the parasite plasma membrane (Fig. 9). How PfSTART1 is delivered to the PVM upon invasion remains to be uncovered. A simple explanation would be that it resides in secretory organelles like dense granules that are released into the developing PV space upon invasion, similar to other PV-resident proteins. Further studies utilising expansion microscopy and lattice light sheet imaging of fluorescently tagged PfSTART1 would help to resolve the localisation of PfSTART1 during and directly following invasion. Coupling these experiments with the now-confirmed PfSTART1 inhibitors would shed further light on the mechanism of action of this antimalarial series and assist in probing the function of this lipid transferase in *Plasmodium*. Antimalarials with new modes of action have never been more urgently needed as marked resistance to artemether-lumefantrine, the most widely used artemisinin combination therapy in Africa, has recently been found in Ugandan clinical isolates[76,77].

## Methods

### *P. falciparum* culture
*Plasmodium falciparum* 3D7 parasites were cultured in human red blood cells provided by the Australian Red Cross Blood Bank, at 4% haematocrit, maintained at 37 °C in a special gas mixture (1% $O_2$, 5% $CO_2$, 94% $N_2$)[78]. The medium used was complete RPMI medium: RPMI-1640 (Sigma), 25 mM HEPES (GIBCO), 0.37 mM hypoxanthine (Sigma), 31.25 µg/mL gentamicin (GIBCO), 0.2% NaHCO$_3$ (Thermo Scientific), 0.5% AlbuMAX II (GIBCO).

### Generating resistance
A clonal population of 3D7 parasites ($10^8$ parasites in five replicates, A to E) were exposed to 10 x EC$_{50}$ = 3 µM of MMV006833 (M-833), until most parasites died. The drug was then removed, and parasites were allowed to recover. Another cycle of drug treatment (3 µM M-833) was then resumed. The resistant lines were cloned out by limiting dilution (diluting the culture into a 96-well plate to achieve an average of ~0.3 parasites per well), prior to growth inhibition assays and genomic DNA (gDNA) extraction (DNeasy Blood and Tissue kit (Qiagen)[18].

### Molecular biology and transfection of *P. falciparum*: Selection Linked Integration and CRISPR/Cas9 methods
Methods for making and transfecting plasmids into *P. falciparum* parasites are described in the Supplementary Information and DNA primer sequences are listed in Table S2.

### Chemistry procedures
Methods for making the M-833 analogues used in this work are outlined in the Supplementary Information.

### Whole genome sequencing and genome reconstruction
Sequencing of M833-resistant parasites and parental 3D7 line was performed as previously described[18]. Genomic sequencing data is available from the European Nucleotide Archive; accession number PRJEB65444.

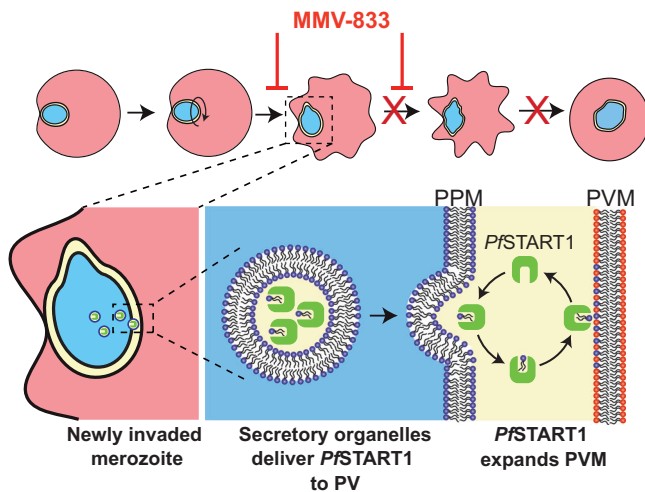

**Fig. 9 | The M-833 series blocks the conversion of newly invaded merozoites into ring-stage parasites by inhibiting the *Pf*START1 protein and reducing the expansion of the parasitophorous vacuole membrane (PVM). (Top)** After invasion, the merozoite begins to spin and produce pseudopodia before differentiating into an amoeboid ring-stage parasite. **(Bottom)** After invasion *Pf*START1, is delivered into the vacuole space possibly via secretory organelles fusing with the parasite plasma membrane (PPM). Here *Pf*START1 could transfer phospholipids from the PPM to the PVM to allow the latter to expand to accommodate the growing ring-stage parasite. The M-833 compound series tightly binds to *Pf*START1, probably displacing the phospholipids and thereby preventing the expansion of the PVM and arresting parasite growth.

## Growth inhibition assays
Parasite growth assays with inhibitory compounds were performed as per[11] and are fully described in the Supplementary Information.

## Egress, invasion & recovery assay
To measure egress, invasion, and follow-up recovery in the presence of different compounds, we adapted the method developed in[11], using Hyp1-Nluc parasites[29]. A full description is in the Supplementary Information.

## Stage arrest and recovery assay
Parasites were synchronised using a Percoll density gradient (Cytiva) combined with 5% sorbitol lysis. Ring-stage parasites at 0-4 hours post-invasion at 2% parasitemia and haematocrit were then added to 2 μM M-833, 60 nM W-991 or 0.02% DMSO and incubated at 37 °C with compound replenished every 24 h. After 48 h, samples of M-833 and W-991-treated parasites were washed x 3 in complete RPMI to remove the compound and put back into culture for the remainder of the experiment. For the following two cycles of growth, samples were taken every 24 h to monitor recovery. Upon completion of time points, fixed cells were stained with 2.5 x SYBR Green (Invitrogen) in PBS, washed once in PBS, and analysed on the Attune Flow Cytometer (ThermoFisher Scientific). Giemsa-stained blood smears were visualised and imaged using a Nikon Eclipse E600 microscope.

## Parasite reduction ratio assay
This was performed as per[78,79] and the full protocol is in the Supplementary Information.

## Lattice light sheet imaging
Parasites were sorbitol synchronised at days 5 and 2 prior to filming. To prepare the parasites for filming, culture was loaded on LS columns attached to MACS MultiStand (Miltenyi Biotec) to isolate late-stage parasites. Imaging medium was prepared by adding 10 μM Trolox (Santa Cruz 53188-07-1) to the culture medium. To compare the effect of drug treatment on parasite invasion, either 60 nM of W-991 or an equivalent DMSO concentration was added to the imaging medium. RBCs were resuspended at 0.5% hematocrit in RPMI-HEPES supplemented with 0.2% sodium bicarbonate and stained with 1.5 μM Di-4-ANEPPDHQ (Invitrogen D36802) membrane marker for 1 h at 37 °C. The stained RBCs were then washed 3 x and resuspended in imaging medium. Purified schizonts were resuspended in culture medium and incubated with 25 nM Mitotracker Deep Red FM (Invitrogen M22426) for 30 min at 37 °C, 5% CO₂. The stained schizonts were then pelleted and the supernatant removed before resuspending the schizonts in the imaging medium. Before imaging, imaging medium with drug was loaded to a well and imaging medium with DMSO control was loaded to another well on an 8-well glass bottom plate (Ibidi 80807). Stained RBCs and stained schizonts were then added to each well and let settle for 30 min. The imaging experiments were performed on Zeiss Lattice Lightsheet 7. 488 nm laser was used to excite Di-4-ANEPPDHQ and 640 nm laser was used to excite Mitotracker Deep Red FM. A quad-notch filter was used to block 405/488/561/640 nm excitation lights. A 290 μm x 200 μm region was scanned with 0.3-0.4 μm interval for both drug-treated and control wells and a simultaneous timelapse was acquired at 2–3 ms exposure time with 15 s interval for 2 h. Acquired data were deskewed and deconvolved using Lattice Lightsheet processing on ZEISS ZEN Blue 3.4 software, then cropped into smaller regions for analyses.

## Vacuole tracking
A data subset was used as training dataset for machine learning on Aivia 10.5.1 software. Single 3D frames were chosen and annotated as either 'Background' or 'Want' on Pixel Classifier analysis tool, where 'Want' is the vacuole area. The Pixel Classifier was applied on other data as feedback for training. Once the trained Pixel Classifier reached satisfactory accuracy, it was applied on the whole dataset in batch mode to obtain 'Want' channel, which is the vacuole confidence map, as additional channel to each file. 3D Object Tracking was then performed based on the 'Want' channel and the vacuoles of interest were isolated from the tracked objects. The sphericity of the vacuoles was extracted from the statistics of the isolated objects. The sphericity data were then plotted on GraphPad Prism 9.5.0 and two-tailed nested t-test was performed on the software to obtain *p*-value between the vacuole sphericity of drug-treated and DMSO control conditions at the initial condition and 15 min after the vacuole is formed.

## Recombinant Expression and Purification
Codon optimised (*Spodoptera frugiperda*, Sf) START domain alone (I149-V394 from PF3D7_0104200) was synthesised by Integrated DNA Technologies and cloned into a modified baculovirus transfer vector (pAcGP67-A) with a N-terminal GP67-signal sequence, 8xHis tag, and TEV protease cleavage site. Recombinant baculovirus was generated via the flashBACTM Baculovirus Expression System using Sf21 cells (Life Technologies), amplified to reach a third passage viral stock, and 1.5% used to infect Sf21 for protein expression. After incubation for three days at 28 °C, media containing secreted protein was harvested by centrifugation and supplemented with 50 mM Tris pH 7.5, 20 mM MgCl₂, 100 mM NaCl, and 10-20 mM imidazole. The supplemented protein sample was purified by nickel affinity chromatography (HisTrap Excel 5 mL, Cytiva), and eluted with 20 mM Tris pH 7.5, 500 mM NaCl, 300-500 mM imidazole. Eluted fractions were further purified using size-exclusion chromatography (Superdex S75 16/600 or Superdex 10/300, Cytiva) with the column pre-equilibrated with 20 mM HEPES pH 7.5, 150 mM NaCl, and peak-containing fractions concentrated and stored at -80 °C until required. If required, prior to ITC, a second size-exclusion chromatography step was performed to remove soluble aggregates.

*S. frugiperda* codon optimised full-length N309K and N330K PfSTART1 mutants were first synthesised by Integrated DNA

Technologies, and then I149 forward and V394 reverse primers utilised to clone the same I149-V394 construct boundaries into the modified pAcGP67a expression vector with a N-terminal GP67-signal sequence, 8xHis tag, and TEV protease cleavage site. As a result of the subcloning, SSG was removed upstream of the TEV cleavage site. Construct boundaries and incorporation of the N309K and N330K mutations was verified by Sanger sequencing. The *Pf*START1 mutants were expressed and purified as per wild-type *Pf*START1.

### *Pf*START1 polyclonal antibody generation and purification

Polyclonal rabbit anti-*Pf*START1 antibodies were generated by the WEHI Antibody Facility using recombinant *Pf*START1 (I149-V394). We further purified the polyclonal antibodies using a *Pf*START1 affinity column that was generated using the AminoLink® Plus Immobilisation Kit (Thermo Scientific), according to the manufacturer's instructions. For this, we immobilised deglycosylated (PNGaseF, NEB) and TEV cleaved *Pf*START1 domain protein that had been further purified by nickel affinity chromatography to remove N-linked glycans and the 8xHis tag. The polyclonal antibody was purified on this matrix according to the manufacturer's instructions.

### Isothermal titration calorimetry

ITC experiments were performed on a MicroCal PEAQ-ITC calorimeter at 25 °C, with a stirring speed of 750 rpm and a reference power of 5 μcal/sec. 500 μM M-833 series inhibitor solutions were diluted to 10 μM in 50 mM $NaPO_4^{-3}$ (pH 7.4), 150 mM NaCl with a final DMSO concentration of 2% (v/v) (cell sample). *Pf*START1 protein was dialysed extensively against 50 mM $NaPO_4^{-3}$ (pH 7.4), 150 mM NaCl, before diluting to 90 μM in the same buffer with 2% (v/v) DMSO (syringe sample). The first injection was 0.4 μL over a 0.8 s duration, and the remaining 19 injections were 2 μL of 4 s duration, with 150 s injection spacing. Data were collected and analysed using the PEAQ-ITC software (MicroCal) and fit by a single site binding model. A fitted offset (constant control heat) was also applied to the integrated heat.

### Solvent profiling Western Blot

3D7 parasites at schizont stage were harvested with 10 x pellet volume of 0.15% saponin in PBS. To extract soluble material, 10 x pellet volume of 0.4% NP40 (IGEPAL CA-630, Sigma-Aldrich) in PBS with 1x Complete Protease cocktail tablet (Sigma-Aldrich) was added to the parasite pellets and 3 cycles of freeze/thawing was performed. Samples were then mechanically sheared by passage through 25 G and 30 G needles before lysate was cleared via centrifugation at 16,000 *g* x 30 min at 4 °C. Soluble fractions were collected and stored at −80 °C until use.

For the solvent challenge, lysate was subjected to either W-991 (10 μM) or DMSO (0.1%) treatment for 3 min before aliquoted into final concentrations of 0-25% of Acetic acid/Ethanol/Formic acid (AEF) at a 50:50:0.1 (v/v/v)[25,26]. The treated lysate and AEF mixture was incubated at 37 °C for 20 min at 800 rpm before aggregates were pelleted at 17,000 *g* x 20 min at 4 °C. Soluble fractions were removed, added to the final concentration of 1x NuPAGE LDS Sample Buffer (Invitrogen) with 1:100 2-mercaptoethanol (Sigma Aldrich) and boiled for 3 min. Proteins were separated on an 4-12% acrylamide gel (NuPAGE, Invitrogen) and subsequently transferred by electroblotting onto nitrocellulose membranes. Blots were probed with primary antibody anti-*Pf*START1 (1:1000), followed by secondary antibody anti-rabbit-HRP (1:4000, Merck Millipore). ECL Plus Western blotting reagent (GE Healthcare) was used to visualize bands with the ChemiDoc Imaging System (Biorad).

### Solvent Proteome Profiling (MS)

The experiment was carried out in three biological replicates. Saponin-liberated mature parasite stages (30-42 hpi) were resuspended in PBS and lysed by 3x flash freeze/thawing using liquid $N_2$, followed by 10 x mechanical sheering with a 29 G needle-syringe, and soluble protein isolation through ultracentrifugation (100,000 g; 20 min, 4 °C). Protein lysate was exposed to the 100 μM of W-991 or the vehicle control (DMSO) for 3 min and subsequently incubated with varying concentration of the solvent mixture 'AEF' (50% Acetone, 50% Ethanol, 0.1% Formate) to a final concentration of 7-25% (v/v) with 2% intervals, for 20 min at 37 °C at 800 rpm on a Thermomixer (Eppendorf). Denatured protein was pelleted through centrifugation (4 °C, 18,000 *g*, 20 min), and the soluble phase was recovered and pulled together in equivolume ratios into two samples; Gradient 1 'G1': 7-15% EAF and Gradient 2 'G2': 17-25% EAF, respectively.

### MS sample preparation

Sample preparation for proteomic analysis was carried out using modified SP4 glass beads protocol[80]. In brief, protein was reduced (20 mM TCEP, 100 mM TEAB) for 20 min at 55 °C and alkylated with 55 mM 2-Chloroacetamide for 30 min, followed by precipitation on beads in 80% ACN with a 6 min centrifugation at 21,000 g and 3 x wash with 80% Ethanol. Dried beads were subjected to sequential digestion with LysC (3 h, 1:50 ratio w/w) and trypsin (overnight, 1:50 ratio w/w), and the resulting digest was acidified with 1% TFA and desalted on T3 C18 stage tips (Affinisep) according to manufacturer's specifications.

### MS data acquisition and data analysis

Peptide samples were analysed on Orbitrap Eclipse Tribrid mass spectrometer that is interfaced with the Neo Vanquish liquid chromatography system. Samples were loaded onto a C18 fused silica column (inner diameter 75 μm, OD 360 μm × 15 cm length, 1.6 μm C18 beads) packed into an emitter tip (IonOpticks) using pressure-controlled loading with a maximum pressure of 1,500 bar, that is interfaced to the mass spectrometer using Easy nLC source and electro sprayed directly into the mass spectrometer. Sample separation was carried out on a linear gradient 5% to 17% of solvent-B at 400 nL /min flow rate (solvent-B: 80% (by vol) acetonitrile) for 23 min and 17% to 25% solvent-B for 10 min, 25% to 34% for 11 min and 34% to 90% solvent-B for 1 min which was maintained at 90% B for 3 min and washed the column at 2% solvent-B for another 3 min comprising a total of 63 min run with a 45 min gradient in a data independent acquisition (DIA) mode MS scan parameters included a full scan using an orbitrap and 120,000 resolution, standard AGC target and maximum injection time of 30 ms. A second DIA experiment used the orbitrap at 30,00 resolution, a precursor scan range of 200-1200 m/z with a normalized AGC target of 3000% and 45 variable isolation windows. Peptides were isolated and fragmented using stepped collision-induced dissociation (HCD) at 24%, 28% and 32% normalized collision energy. Peptide identification was carried out in DIA-NN 1.8.1 using standard settings and an in silico spectral library generated from Uniprot *P. falciparum* (UP000001450) and human (UP000005640) reference proteomes. The peptide length range was set to 7-30 amino acids. One missed cleavage and 1 variable modification were allowed (ox(M) and Ac(N-term)). Precursor FDR was set to 1% and the match between runs was on, precursor charge range was set between 1-4, precursor m/z range of 300−1800, fragment ion m/z range of 200-1800. Subsequently, the differential abundance analysis (moderated t-test, based on limma package[81]) of *P. falciparum* proteins was conducted in the R environment (R version 4.2.0; R Studio Version 2023.12.1 + 402) using precursor normalised MaxLFQ data for proteins detected with ≥2 peptides. Hit selection criteria included p value < 0.01, log2 fold change of >0.73 in protein abundance and protein detection across all tested samples.

### Gametocyte culturing and standard membrane feeding assays

These assays were performed at the Walter and Eliza Hall Institute and at the London School of Hygiene and Tropical Medicine using slightly different approaches which are described in the Supplementary Information.

## Generation of *Plasmodium berghei* sporozoites

*P. berghei* ANKA constitutively expressing mCherry[82] was used for the in vitro liver stage invasion assay. Animals used for the generation of the sporozoites were 4- to 5-week-old male Swiss Webster mice and were purchased from the Monash Animal Services (Melbourne, Victoria, Australia) and housed at 22 to 25 °C on a 12 h light/dark cycle with 40-70% humidity at the School of Biosciences, The University of Melbourne, Australia. All animal experiments were in accordance with the Prevention of Cruelty to Animals Act 1986, the Prevention of Cruelty to Animals Regulations 2008, and National Health and Medical Research Council (2013) Australian code for the care and use of animals for scientific purposes. These experiments were reviewed and permitted by the Melbourne University Animal Ethics Committee (2015123).

Infections of naïve Swiss mice were carried out by intraperitoneal (IP) inoculation obtained from a donor mouse between the first and fourth passages from cryopreserved stock. Parasitemia was monitored by Giemsa smear and exflagellation quantified 3 days post-infection. *A. stephensi* mosquitoes were allowed to feed on anaesthetised mice once the exflagellation rate was assessed between about 12 to 15 exflagellation events per $1 \times 10^4$ RBCs. Salivary glands of infected mosquitoes (days 17 to 24 post-infection) were isolated by dissection and parasites placed into RPMI-1640 media.

## In vitro liver invasion assays

This was performed essentially as described[83] with the minor variations outlined in the Supplementary Information.

## Dual gamete formation assays

The compounds were tested in the *P. falciparum* Dual Gamete Formation Assay (*Pf*DGFA)[84], fully described in the Supplementary Information.

## Protein extraction & Western Blot

Parasites were prepared for the stage of interest and proteins were extracted using a saponin-lysis (unless indicated otherwise). For knock-down assays, parasites were exposed to 0 or 2.5 mM GlcN for 48 h (starting from late schizont/early rings): the day before harvesting schizonts, 30 nM ML10 (LifeArc) was also added to the cultures to block egress. Infected red blood cells were lysed 10 min on ice with 0.1% saponin in PBS complemented with 1x Protease Inhibitors Cocktail (Roche; PBS + PI). Parasites were pelleted and washed with PBS + PI to remove haemoglobin. The pellets were resuspended in 10 to 20 x volume of non-reducing sample buffer (NRSB; 50 mM Tris-HCl pH 6.8, 2 mM EDTA, 2% SDS, 10% glycerol, 0.005% phenol blue), sonicated a minimum of 3 × 30 seconds (Diagenode sonicator), were optionally reduced with 100 mM dithiothreitol, and boiled 10 min at 80 °C. Protein samples were centrifuged at 15,000 g, then run on a pre-cast 4-12% NuPAGE Bis-Tris gel (Invitrogen) and proteins were transferred onto a nitrocellulose membrane using iBlot (Invitrogen). Membranes were exposed to the primary antibody overnight at 4 °C, to the secondary antibody 1 h at room temperature, and fluorescence was measured using an Odyssey imaging system, which was also used to measure densitometry. The list of antibodies used, their dilution, and origins are described in Table S3.

## Processing of *Pf*START1 along schizogony

Magnet-purified schizonts were harvested after a 4 h treatment with the following: DMSO (0.1%), 49c (10 nM), C1 (1.5 µM) or E64 (10 µM). Parasites were washed in ice-cold PBS + PI (centrifugation steps carried out at 4 °C at 3,000 g). Samples were resuspended in NRSB, sonicated, boiled 10 min at 80 °C and separated by electrophoresis as explained previously. The parasites used in this experiment were 3D7 transfected with p1.2-ABH(WT)-83-HA-*glm*S, i.e. parasites containing a recodonised version of PF3D7_0403800 within its genome.

## Differential lysis assay

Schizont-infected RBCs were sequentially lysed with (1) equinatoxin II (EqtII), which lyses the RBC membrane, (2) saponin, which lyses the PVM, (3) TX-100, which lyses all membranes. All the lysis buffers were made in PBS + PI, supernatants were collected in fresh tubes, and pellets were washed twice in PBS + PI. ML10-treated schizonts were enriched (with Percoll) and lysed in 10 x pellet volume of EqtII (at a concentration empirically determined to obtain 100% hemolysis) for 10 min at 37 °C. Lysate was centrifuged at 1000 g for 5 min to collect the supernatant. The pellet was washed and then lysed in 10 x volume of 0.03% saponin (10 min on ice, centrifuged at 16,000 g for 1 min at 4 °C). The supernatant was collected, the pellet was washed and lysed in 10 x volume of 0.25% TX-100 (10 min on ice, centrifuged at 16,000 g for 1 min at 4 °C, and the supernatant collected). All supernatants were mixed with NRSB, boiled 10 min at 80 °C, and separated by electrophoresis as explained previously.

## Carbonate extraction and Proteinase K protection assay

These were conducted as previously described[38,85] with full details in the Supplementary Information.

## Reporting summary

Further information on research design is available in the Nature Portfolio Reporting Summary linked to this article.

## Data availability

Genomic sequencing data is available from the European Nucleotide Archive; accession number PRJEB65444. Mass spectrometry data is available from JPOST Repository; accession number PXD048262 [https://repository.jpostdb.org/entry/JPST002439.1]. Source data are provided in this paper.

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

## Acknowledgements

We acknowledge the traditional custodians of the lands on which this project was conducted: the Wurundjeri and the Boon Wurrung people of the Kulin nation. We thank Lifeblood Biological Resources Australia for providing the human red blood cells and LifeArc for supplying ML10. We thank Danu Marapana for kindly sharing p1.2 CRISPR plasmid. We thank the WEHI screening facility for conducting parasite growth assays with newly synthesised compounds. This work was supported by the Victorian Operational Infrastructure Support Program received by the Walter and Eliza Hall and Burnet Institutes. This work was funded by the National Health and Medical Research Council of Australia (Ideas Grant to W.N. and P.G. 2001073; Development Grant 1135421 to B.E.S. and A.F.C.; Ideas Grant to K.L.R and N.D.G 2012271). A.F.C. is a Howard Hughes International Scholar and an Australia Fellow of the NHMRC. B.E.S. is a Corin Centenary Fellow. J.M.D. is a Human Frontier Science Program Fellow. MTF is supported by a grant from the Medicines for

Malaria Venture (RD-21-2003) awarded to MJD. MJD is supported by a UKRI Medical Research Council Career Development Award (MR/V010034/1). Mass Spectrometry sample analysis was supported by WEHI Proteomics Facility. We thank Kirsty McCann for lending expertise on bioinformatic analyses of the resistant lines.

## Author contributions

Study design and planning: M.G.D., C.B., G.M.W., W.N., J.M.D., K.R., N.D.G., C.D.G., W-H.T., B.E.S. and P.R.G. Performed experiments and generated reagents: M.G.D., C.B., G.M.W., W.N., J.M.D., C.E., Z.R., M.J.M., C.D.G., D.B.L., T.K.J., J.T., M.T.F., M.K., K.R., H.P., L.B.S., L.B-G. Data analysis: M.G.D., C.B., G.M.W., W.N., J.M.D., K.R., S.M., N.D.G., C.D.G., C.J.S., M.J.D. and H.P. Provided funding and supervision: G.I.M., A.E.B., B.S.C., T.F.dK-W., K.L.R, A.F.C., and P.R.G. Manuscript writing: M.G.D., C.B., and P.R.G. with contributions from other authors.

## Competing interests

The authors declare no competing interests.

## Additional information

[1]Burnet Institute, Melbourne, VIC 3004, Australia. [2]Walter and Eliza Hall Institute, Parkville, VIC 3052, Australia. [3]Institute of Mental and Physical Health and Clinical Translation (IMPACT) and School of Medicine, Deakin University, Geelong, VIC 3220, Australia. [4]Department of Medical Biology, The University of Melbourne, Parkville, VIC 3010, Australia. [5]School of Biosciences, The University of Melbourne, Parkville, VIC 3010, Australia. [6]Department of Microbiology and Immunology, The University of Melbourne, Parkville, VIC 3010, Australia. [7]Department of Molecular Biology, Umeå University, Umeå 901 87, Sweden. [8]The Laboratory for Molecular Infection Medicine Sweden (MIMS), Umeå, Sweden. [9]Department of Infection Biology, Faculty of Infectious Diseases, London School of Hygiene and Tropical Medicine, WC1E 7HT London, UK. [10]Wellcome Trust Human Malaria Transmission Facility, Faculty of Infectious & Tropical Diseases, London School of Hygiene & Tropical Medicine, London WC1E 7HT, UK. [11]Monash University, 3800 Melbourne, VIC, Australia. [12]Present address: Department of Microbiology and Molecular Medicine, University of Geneva, Geneva 1206, Switzerland. [13]These authors contributed equally: Madeline G. Dans, Coralie Boulet. ✉e-mail: dans.m@wehi.edu.au; paul.gilson@burnet.edu.au

