## [Peer Review File · Nature Communications]

Aryl amino acetamides prevent Plasmodium falciparum ring development via targeting the lipid-transfer protein PfSTART1Reviewers' Comments:

Reviewer #1:

Remarks to the Author:

This article critically examines the mode of action of the compound M-833 against *Plasmodium falciparum*. Dans et al. and colleagues identified PfSTART1 as the molecular target of M-833, employing a drug pressure assay alongside the novel Solvent PISA technique, which they introduced for the first time in their study. Their findings suggest that M-833, alongside its more potent derivatives like W-991, holds promise in inhibiting *P. falciparum* at various lifecycle stages (blood stage invasion of erythrocytes) and could potentially block the parasite's transmission to mosquitoes. They did not have an effect on the liver stage (done with *P. berghei*, not *P. falciparum*)
The drug pressure with M-833T pinpointed mutations within the PfSTART1 gene—N309, I224F, and N330K—located in the central domain of PfSTART1 . By integrating these mutations into the genome and comparing them to the wild type (wt) as a control, the study demonstrated these specific mutations are responsible for resistance.

I suggest integrating Figure 2 and S2 to coalesce drug assay results with western blot analyses for a more improved presentation of the results. The study's narrative on the introduction of compound 2 and the subsequent creation of W-991 effectively illustrates the structural similarities between these compounds and M-833, implying they target the same molecular sites. However, it's crucial also to explore the possibility of cross-resistance mechanisms and genetic linkage before it is shown that two compounds have the same target. This line of inquiry led to the synthesis of compound W-991, notable for its significant, 20-fold enhanced activity compared to M-833. The mention of compound 3 on page 10 requires additional details to understand its contribution to the study entirely.

Page 11 and 12, it is unclear if the SIPP and PISA assay was done with M-833 or W-991.

Interestingly, M-833 and W-991 do not impact the liver stage but do affect parasite transmission (shown for W-991) without affecting the gametocyte development.

The most exciting part is the processing of PfSTART1 during the schizont stage. There is an increase in processed forms of PfSTART from 40 to 48 h post-invasion.

Figure 7 Aii should be labeled PfSTART1; number 1 is missing.

Overall, this article highlights the innovative approach taken by Dans et al. and colleagues to elucidate the action of M-833, a compound with improved activity *P. falciparum*, emphasizing the potential for new therapeutic strategies in combating malaria. The processing of the PfSTART1 is the highlight of the paper.

Reviewer #2:

Remarks to the Author:

This study by Dans et al identifies PF37_0104200 as a target of aryl acetamides previously shown to inhibit *P. falciparum* development at the early ring stage. PF37_0104200 (here called PfSTART1 but also previously referred to as PV6 by others), which encodes a START domain and localizes to the parasite vacuole, has been shown to transport several phospholipids and to be essential for ring stage parasite development. The study is divided into two main sections: in the first, the authors evolve parasite resistance to M-833 and identify two mutations in PfSTART1 that confer resistance in engineered isogenic mutant lines. Selection linked integration was used initially to introduce the mutations; interestingly, the fusion with the P2A skip peptide-linked selection cassette resulted in hypersensitivity to M-833 in the WT control line, prompting the authors to regenerate these mutants using CRISPR/Cas9. This provided further confirmation that N309K and N330K confer resistance but

also suggests that modulation of the pool of PfSTART1 in the SLI-WT parasites (likely by inefficient P2A skipping that reduces the level of functional WT protein) sensitizes parasites. In agreement with this, knockdown of PfSTART1 also results in hypersensitivity and similar results were seen for W-991, an optimized M-833 derivative with increased potency. Structural prediction indicates N309 and N330 are in the lipid binding pocket of the START domain and ITC showed both compounds bound recombinant PfSTART1 but not the N330K mutant. Furthermore, a SIPP assay identified W-991 interaction with PfSTART1 from parasite lysates and PfSTART1 was among 4 proteins identified to interact with W-991 by solvent PISA. Collectively, these data convincingly demonstrate the compounds bind PfSTART1 and loss of interaction in the identified mutants confers resistance. This is an important finding as anti-malarials with novel modes of action are needed.

The second section explores compound activity on parasite stages as well as the biology and biochemistry of PfSTART1. These later data recapitulate some previous observations but also go further with lattice light sheet microscopy analysis of ring morphology and the possible identification of an additional mature species that comprises a minority of the PfSTART1 pool. The study is well written and the data are robust – I especially appreciated the use of several orthologous approaches to validate the interaction between the compounds and PfSTART1, including the solvent PISA approach which will likely be an exciting new tool for anti-malarial target discovery.

Major Comments:

-The clones isolated from the M-833 selections appear to be less resistant than the isogenic mutant lines (Fig 3B,C). Is there any change in the expression level of PfSTART1 in PopD-D7 or PopE-F10 compared to the parental line that could explain this? Do the data in Fig S2D and S3D show any change in basal PfSTART1 expression relative to 3D7? The Solvent-PISA data also identified interactions with MAAP, RESA3 and SRP19 - were mutations in these genes identified in any of the selected populations that might also be contributing?

-Is the fitness defect in the selected resistant lines (Fig 1D) a result of compromised PfSTART1 function? In other words, do the isogenic PfSTART1 N309K and N330K mutants show the same fitness defect as the selected lines or is this a result of other changes? Also, do the isogenic mutant parasites undergo normal ameboid transition (in the presence or absence of the compounds)?

-The data convincingly show compound binding to PfSTART1 and that loss of this interaction leads to resistance. The authors conclude that M-833 and analogs inhibit PfSTART1 activity (in the title and other places), however PfSTART1 activity (lipid transport activity) is not assessed. This assay (previously performed by other groups using recombinant PfSTART1) would validate this conclusion; alternatively, in the absence of an activity inhibition assay, this conclusion should be re-worded.

-Fig S6A: While the 12 hr (ring) and 24 hr (troph) parasites treated with M-833 or W-991 look similar to the DMSO control, the 36 hr schizonts appear quite different to me. What is the basis for the claim that late stage parasite development is not impacted when treatment is initiated in rings? Do parasites treated in the first cycle produce equivalent numbers of new rings in the second cycle as the control?

-The authors claim that the inhibitors have no activity on the liver stage but the data in Fig S8A is only measuring *P. berghei* sporozoite invasion of HC-04 cells, not liver stage development. The conclusion should be changed to no impact on sporozoite invasion. Additionally, the authors need to show that their compounds are also effective against the *P. berghei* blood stage to be able to interpret this result.

Minor Comments:

-Can an internal population of PfSTART1 be detected by IFA that might correspond with the lower molecular weight PfSTART1 species?

-Is a higher molecular weight band observed outside the cropped area in the blot in Fig S2D that would correspond to a population that did not skip at P2A? If P2A is not skipping efficiently in this fusion, it would strengthen the idea of a reduced functional pool of PfSTART1.

-In the title to the lattice light sheet microscopy section, "membrane disruption" is perhaps misleading. The data show that differentiation into normal ring morphology does not occur while membrane disruption suggests the membrane has been compromised or lost.

-For clarity, please indicate that the cyan is mitotracker (which marks the merozoite) in the Fig 6 legend as in the legends to the Supplementary movies.

-What do the marks on the Y axis (sphericity) represent in Fig S7A?

-SLI-WT PfSTART1 migration is quite different between Fig S2D (~50 kDa) and Fig 7A (~60 kDa – although a second marker is needed to gauge this). Are the marker labels correct? It would be helpful if at least two markers could be included in each cropped blot so that the sizes can be estimated (or include the uncropped blots).

-The proteinase protection experiments are difficult to interpret since there is degradation in both the PBS and SAP +protK lanes and no density quantification was provided to back up the claims (ie, PTEX150 was most strongly degraded in sap (see below)). Also, in the replicate in Fig S9E, there is little to no signal in the PBS -protK lane? This prevents comparison of the level of degradation between the PBS and SAP treatments but isn't commented on.

"PfGBP130 was localised in the RBC as it was degraded in the EqtII supernatant" - I'm confused by this statement since there is no protK addition to the EqtII supernatant (left most lane in Fig 7E). Perhaps the authors mean "released" instead of "degraded"?

"PfPTEX150 was most strongly degraded upon addition of saponin indicating it was mainly located in the PV." – but PTEX150 shows even more degradation in the Tx100 treatment than the Saponin treatment?

"whilst PfSTART1proc1 and PfSTART1proc2 could reside within the parasite and protected from proteinase K cleavage." - although I agree the proc 1 band is lost in the Tx100 + protK lane, the proc 2 band is hard to interpret (relative to the Tx100 -protK lane) because of the apparent doublet of degradation products migrating in the same area.

-What are the lower bands partly cropped out in the GBP130 blot in replicate proteinase protection assay in Fig S9E?

-Do the changes in the SEC elution profile of recombinant PfSTART1 between WT and N309K (Fig S5A) provide any clue to how this mutation might be impacting the protein?

-A "be" should be inserted before "protected" in the sentence: "...could reside within the parasite and protected from proteinase K cleavage."

-“Sensitize” should be “sensitivity” in the sentence: “...could interfere with PfSTART1 function and sensitize of SLI-WT parasites to M-833”.

-Cut “intervals” or change “every” to “at” in the sentence: “...proteins of tightly synchronized wildtype 3D7 schizonts without the HA tag were analysed every 4 h intervals”

REVIEWER COMMENTS

We thank the reviewers for their comments and insightful suggestions. Our responses to their queries are in red text and changes to in-text and in red italics.

Reviewer #1 (Remarks to the Author):

This article critically examines the mode of action of the compound M-833 against *Plasmodium falciparum*. Dans et al. and colleagues identified PfSTART1 as the molecular target of M-833, employing a drug pressure assay alongside the novel Solvent PISA technique, which they introduced for the first time in their study. Their findings suggest that M-833, alongside its more potent derivatives like W-991, holds promise in inhibiting *P. falciparum* at various lifecycle stages (blood stage invasion of erythrocytes) and could potentially block the parasite's transmission to mosquitoes. They did not have an effect on the liver stage (done with *P. berghei*, not *P. falciparum*)

The drug pressure with M-833T pinpointed mutations within the PfSTART1 gene—N309, I224F, and N330K—located in the central domain of PfSTART1. By integrating these mutations into the genome and comparing them to the wild type (wt) as a control, the study demonstrated these specific mutations are responsible for resistance.

Comment: I suggest integrating Figure 2 and S2 to coalesce drug assay results with western blot analyses for a more improved presentation of the results.

Response: We thank the reviewer for this suggestion. For improved presentation of the results, we split Fig2 into two: the new Fig2 incorporates the construction map of the SLI and CRISPR parasites together with the initial growth assay on those parasites, while the new Fig3 contains, as suggested, the western blots and densitometry showing the efficiency of PfSTART1 knock-down, together with the EC50 with/without knock-down. Supplementary figures 2 and 3 were combined: new FigS2 contains the PCR analysis (agarose gel and sequencing) of the clonal lines, together with the parasites' growth in the absence of drug (with or without glucosamine).

The study's narrative on the introduction of compound 2 and the subsequent creation of W-991 effectively illustrates the structural similarities between these compounds and M-833, implying they target the same molecular sites. However, it's crucial also to explore the possibility of cross-resistance mechanisms and genetic linkage before it is shown that two compounds have the same target. This line of inquiry led to the synthesis of compound W-991, notable for its significant, 20-fold enhanced activity compared to M-833. The mention of compound 3 on page 10 requires additional details to understand its contribution to the study entirely.

Response: Upon reflection we agree that this was not clear. We currently have a manuscript under review at European Journal for Medicinal Chemistry (Nguyen et al. 2024) that describes the structure activity relationship of this chemical series in detail. However, to aid in readability for this current manuscript, we have updated the figure to show compound 3 as the “intermediate hybrid” in generating the potent analogue W-991 (Fig3A, now Fig4A). We have also updated the text as follows:

*“Combining the gem dimethyl and the aryl oxy structural elements from M-833 and compound 2 respectively, and removal of the 2-methyl group from M-833 led to the intermediate hybrid compound 3 that exhibited a 10-fold improvement in antiparasitic activity (EC50 = 120 nM) relative to the activity of M-833. Replacing the aryl oxy functionality in compound 3 with an amino group led to **W-991** (WEHI-991), resulting in a 20-fold improvement in parasite activity (EC50 = 7 nM) (Fig 4A).”*

Comment: Page 11 and 12, it is unclear if the SIPP and PISa assay was done with M-833 or w-991.

Response: We direct the reviewer to the in-text referral to W-991 in the SIPP experiments: *“lysate was exposed to increasing concentrations (0-25%) of a mixture of acetone, ethanol and formic acid (AEF) in the presence of W-991 or DMSO”*. In addition to the labels of ‘W-991’ in the associated Figure 5.

Interestingly, M-833 and W-991 do not impact the liver stage but do affect parasite transmission (showed for W-991) without affecting the gametocyte development.

Response: We also agree that this was an unexpected result. We thank the reviewer for pointing this out as it has made us reflect that only sporozoite invasion activity was tested in the presence of the PfSTART1 inhibitor, not liver stage development. To make this clearer in the text we have changed the results heading to:

“PfSTART1 inhibitors do not prevent sporozoite invasion but block parasite transmission to mosquitoes.”

We have also added the following paragraph to the discussion to highlight this point and future work required to test the inhibitors against liver stage development:

“The activity profile of the PfSTART1 inhibitors was found to be variable in other stages of the lifecycle outside the blood stage with no drug effect against liver stage invasion, stage V gametocyte development and gametogenesis. The lack of activity against liver stage of infection was surprising given that sporozoites grow and replicate in a PV, similar to the asexual blood stage^{60,61}. The apparent inactivity of the PfSTART1 inhibitors could be due to the experimental design where, sporozoites and W-991 were added to human liver cells and

evaluated only 2 h post invasion and it is possible we did not capture the developmental inhibition of the PV formation. Further experiments where sporozoites are added to liver cells with the compounds for a longer period are required to definitively rule out liver activity of this series."

The most exciting part is the processing of PfSTART1 during the schizont stage. There is an increase in processed forms of PfSTART from 40 to 48 h post-invasion.

Comment: Figure 7 Aii should be labeled PfSTART1; number 1 is missing.

Response: We thank the reviewer for spotting this omission: Figure 7 Aii (now Fig8B) was amended to state "PfSTART1".

Overall, this article highlights the innovative approach taken by Dans et al. and colleagues to elucidate the action of M-833, a compound with improved activity *P. falciparum*, emphasizing the potential for new therapeutic strategies in combating malaria. The processing of the PfSTART1 is the highlight of the paper.

Reviewer #2 (Remarks to the Author):

This study by Dans et al identifies PF37_0104200 as a target of aryl acetamides previously shown to inhibit *P. falciparum* development at the early ring stage. PF37_0104200 (here called PfSTART1 but also previously referred to as PV6 by others), which encodes a START domain and localizes to the parasite vacuole, has been shown to transport several phospholipids and to be essential for ring stage parasite development. The study is divided into two main sections: in the first, the authors evolve parasite resistance to M-833 and identify two mutations in PfSTART1 that confer resistance in engineered isogenic mutant lines. Selection linked integration was used initially to introduce the mutations; interestingly, the fusion with the P2A skip peptide-linked selection cassette resulted in hypersensitivity to M-833 in the WT control line, prompting the authors to regenerate these mutants using CRISPR/Cas9. This provided further confirmation that N309K and N330K confer resistance but also suggests that modulation of the pool of PfSTART1 in the SLI-WT parasites (likely by inefficient P2A skipping that reduces the level of functional WT protein) sensitizes parasites. In agreement with this, knockdown of PfSTART1 also results in hypersensitivity and similar results were seen for W-991, an optimized M-833 derivative with increased potency. Structural prediction indicates N309 and N330 are in the lipid binding pocket of the START domain and ITC showed both compounds bound recombinant PfSTART1 but not the N330K mutant. Furthermore, a SIPP assay identified W-991 interaction with PfSTART1 from parasite

lysates and PfSTART1 was among 4 proteins identified to interact with W-991 by solvent PISA. Collectively, these data convincingly demonstrate the compounds bind PfSTART1 and loss of interaction in the identified mutants confers resistance. This is an important finding as anti-malarials with novel modes of action are needed.

The second section explores compound activity on parasite stages as well as the biology and biochemistry of PfSTART1. These later data recapitulate some previous observations but also go further with lattice light sheet microscopy analysis of ring morphology and the possible identification of an additional mature species that comprises a minority of the PfSTART1 pool. The study is well written and the data are robust – I especially appreciated the use of several orthologous approaches to validate the interaction between the compounds and PfSTART1, including the solvent PISA approach which will likely be an exciting new tool for anti-malarial target discovery.

Major Comments:

Comment:-The clones isolated from the M-833 selections appear to be less resistant than the isogenic mutant lines (Fig 3B,C). Is there any change in the expression level of PfSTART1 in PopD-D7 or PopE-F10 compared to the parental line that could explain this? Do the data in Fig S2D and S3D show any change in basal PfSTART1 expression relative to 3D7? The Solvent-PISA data also identified interactions with MAAP, RESA3 and SRP19 - were mutations in these genes identified in any of the selected populations that might also be contributing?

Response:

- (1) "The clones isolated from the M-833 selections appear to be less resistant than the isogenic mutant lines (Fig 3B,C)."

We thank the reviewer for bringing this to our attention but we think the differences between Fig4B and 4C (previously 3B and 3C) namely, the EC₅₀ values between resistant clones (PopD-D7, PopE-F10) and isogenic mutants (CR-N309K and CR-N330K) are not really that great.

Our data show that, EC₅₀(W991) on PopD-D7 and PopE-F7 is > 200 nM (exact value not determined in this particular assay; Fig4B), and EC₅₀(W991) on CR-N309K and CR-N330K mutants is 687 and 623 nM respectively (Fig4C).

Regarding M-833, it is a bit more difficult to compare directly the EC₅₀ values between the resistant clones and the CRISPR constructs because M-833 of different origins were used

(commercially purchased vs house made), with slightly different EC₅₀ on 3D7 parasites. Noting that the M-833 batch used in Fig2D (previously Fig2B) provided a lower EC₅₀ on 3D7 than the batch used in Fig4B (previously Fig3B), we would argue that the EC₅₀ values for resistant clones compared to CRISPR constructs are probably quite similar.

Find a summary table of the EC₅₀ values below:

EC ₅₀ (from which figure)	PopD-D7	CR-N309K	PopE-F10	CR-N330K
W-991	> 200 nM (Fig4B, prev. Fig3B)	687 nM (Fig4C, prev. Fig3C)	> 200 nM (Fig4B, prev. Fig3B)	623 nM (Fig4C, prev. Fig3C)
M-833* (different batches of compound)	4.88 μM (Fig4B, prev. Fig3B)	4 μM (Fig2D, prev. Fig2B)	> 10 μM (Fig4B, prev. Fig3B)	9.5 μM (Fig2D, prev. Fig2B)

In contrast to the CR-mutants, the SLI mutants were less resistant than the original resistant clones (also see responses to other questions below), which we attributed to the additional HA tag and skip peptide P2A on the C-terminus of the protein.

- (2) Is there any change in the expression level of PfSTART1 in PopD-D7 or PopE-F10 compared to the parental line that could explain this? Do the data in Fig S2D and S3D show any change in basal PfSTART1 expression relative to 3D7?

We thank the reviewer for this excellent question. Unfortunately, we do not possess the data at this time regarding expression levels of *PfSTART1* in PopD-D7 and PopE-F10 parasites. Regarding SLI and CRISPR parasites, you can find the expression levels (compared to 3D7) below. It seems that all the SLI parasites have less *PfSTART1* compared to 3D7, but CRISPR parasites seem to have similar levels to 3D7. The low levels of *PfSTART1* in SLI parasites is very intriguing. It might be contributing to their increased sensitivity to M-833 (similar to how parasites are sensitized upon *PfSTART1* knock-down). It would be very interesting to further investigate why SLI parasites appear to contain less *PfSTART1* (diminished expression, impeded translation, increased degradation?). In any case however, it does not appear that the mutants express different levels of *PfSTART1* compared to their WT controls in either the SLI or the CRISPR constructs.

Expression levels of PfSTART1 in 3D7, SLI- and CRISPR- parasites normalized by PfEXP2 (left graphs) or PfHSP70.1 (right graphs). Densitometry from western blots depicting knock-down upon addition of glucosamine (only the condition with 0 mM GlcN was measured).

(3) The Solvent-PISA data also identified interactions with MAAP, RESA3 and SRP19 - were mutations in these genes identified in any of the selected populations that might also be contributing?

There was no mutation identified in these genes in the clones obtained from M-833-resistance selection (see Table S1). We think that binding of W-991 to MAAP, and perhaps to RESA3 and SRP19 (their *p* values were above, but very close to the significance threshold) may occur without contributing to the mode of action. When comparing the structures of PfSTART1 and PfMAAP (PF3D7_1035900) in Alphafold, there were no obvious structural similarities to PfSTART1.

We also fully reproduced resistance to the M-833 series when we generated our CRISPR mutants, indicating that there are no secondary mutations that are contributing to the resistance. We have added the following sentence in the Discussion:

"Of note, the other proteins stabilised in the Solvent PISA assay (PfMAAP, PfRESA3 and PfSRP19) did not contain any mutations in the initial resistance selection suggesting protein binding to W-991 occurs without contributing to the mode of action of the compound."

Comment: Is the fitness defect in the selected resistant lines (Fig 1D) a result of compromised PfSTART1 function? In other words, do the isogenic PfSTART1 N309K and N330K mutants show the same fitness defect as the selected lines or is this a result of other changes? Also, do the isogenic mutant parasites undergo normal amoeboid transition (in the presence or absence of the compounds)?

Response: We assume the drug-selected N309K and N330K parasites grow more slowly due to impaired lipid transferase activity but cannot prove this because we have not yet developed a lipid transferase assay for *PfSTART1*. To further investigate this, we examined some data we had on the engineered mutants.

All the SLI parasites (including SLI-WT) grew less well than 3D7 possibly due to the C-terminal addition of the HA epitope/P2A skip peptide and reduced expression (discussed in another question). SLI-N330K but not SLI-N309K parasites, grew more slowly than their SLI-WT control (see graph below). In contrast, both drug-selected mutants grew more slowly than their respective 3D7 control. Why the engineered mutants behaved differently than the drug-selected mutants is not known.

Next, we measured the growth rates of the CRISPR parasites lines and found that that CR-N309K and CR-N330K grew as well as the CR-WT control (see graph below). Since we do not have a solid explanation for why the original drug-selected mutants grew more slowly than their 3D7 control in contrast to the CR-N309K and CR-N330K mutants that grew the same their CR-WT control, we will remove the fitness cost experiment (Fig 1D and associated text) from the manuscript. This issue is going to require further investigation perhaps once lipid transferase assays have been developed. We thank the reviewer for bringing this to our attention.

Regarding the transition of mutant parasites to amoeboid rings: due to the significant effort involved we did not analyse the mutant parasites by lattice light sheet microscopy to observe their development. However, during culturing of these lines (the original resistant clones, the SLI and CRISPR lines), we did not observe any significant delays in ring development by Giemsa smears which would be indicative of impairment of normal amoeboid transition.

Growth of synchronized parasites was measured over several cycles, every 48 h (at trophozoite stage). Top panel- original resistant lines and SLI transgenic lines. Bottom panel- CRISPR transgenic lines. Statistical test: mixed-effects analysis, Tukey's multiple comparison tests.

-The data convincingly show compound binding to PfSTART1 and that loss of this interaction leads to resistance. The authors conclude that M-833 and analogs inhibit PfSTART1 activity (in the title and other places), however PfSTART1 activity (lipid transport activity) is not assessed. This assay (previously performed by other groups using recombinant PfSTART1) would validate this conclusion; alternatively, in the absence of an activity inhibition assay, this conclusion should be re-worded.

Response: We agree with this statement as the reviewer rightfully has pointed out we have not shown lipid binding activity is altered in the presence of the *PfSTART1* inhibitors. To address this, we have changed the title of the manuscript to reflect their target of *PfSTART1* rather than their effect on lipid transfer activity:

"Aryl amino acetamides prevent Plasmodium falciparum rings development via targeting the lipid-transfer protein PfSTART1"

Where appropriate, we have also changed referring to the compound as *PfSTART1* "inhibitors" as realized by the following in-text changes:

"The PfSTART1-targeting compounds inhibitors also blocked transmission..." (abstract)

"We further investigate how the M-833 series bind to PfSTART1 and how this blocks merozoite development into ring-stage parasites."

"The optimised analogue, W-991, had improved binding affinity for PfSTART1..."

"... ITC and Solvent-PISA experiments strongly support PfSTART1 as the principal molecular target of the M-833 series"

"To investigate whether parasites could recover from M-833 and W-991 treatment..."

We now also explain our reasoning behind referring to the compounds as "PfSTART1 inhibitors" as follows:

*"A short 4 h treatment was also sufficient to affect the morphology of the treated parasites for days: at 72 h, M-833 and W-991-treated parasites resembled the dysmorphic ring-stage parasites described previously in the genetic knockout of PfSTART1 (Fig 6A)^{11,17}. **Based on the phenotype reminiscent of the PfSTART1-knockout and the data shown in the paper, we are hereafter referring to M-833 and W-991 as PfSTART1 inhibitors.**"*

In the discussion, we have now added the following sentence to make it clear to the reader we require future work to conclude the compound series is inhibiting PfSTART1 activity:

"Overall, this demonstrates that M-833 and analogues exert their antiparasitic activity by targeting PfSTART1. The data presented here strongly suggest that the compounds studied inhibit PfSTART1's lipid transfer activity, but it will be important in the future to confirm this using specific transferase activity assays."

-Fig S6A: While the 12 hr (ring) and 24 hr (troph) parasites treated with M-833 or W-991 look similar to the DMSO control, the 36 hr schizonts appear quite different to me. What is the basis for the claim that late stage parasite development is not impacted when treatment is initiated in rings? Do parasites treated in the first cycle produce equivalent numbers of new rings in the second cycle as the control?

Response: We thank the reviewer for this astute observation which encouraged us to re-examine our data. For FigS6A (now FigS5A), we turned to our corresponding FACS data for

quantification of parasitemia. As seen below, this showed an approximate 1.5-fold reduction in parasitemia with the *Pf*START1 inhibitors at the 48-60 h timepoint which corresponds to the differences in schizont maturity the reviewer pointed out.

This made us re-examine and quantify the stages in our Giemsa-stained blood smears from this experiment. Here there is also a slight reduction in parasitemia upon drug treatment in the 48 h time point. However, once we quantified the 60 h time point in the smears, M-833 treatment had essentially “caught up” with similar parasitemia to DMSO (albeit deformed rings compared to healthy trophozoites in the DMSO treatment), whilst W-991 exhibited a slight 1.3-fold reduction in parasitemia. Due to low DNA content, Sybr green staining would not be able to differentiate these deformed rings so examining the smears in this case was a more accurate representation of stage-specific parasitemia.

In Dans et al 2020, we show that when M-833 was added to schizonts and merozoite invasion was tracked by live cell imaging, invasion itself was slowed down but not reduced with the same number of invasions per egress as DMSO. It therefore appears there is a slight developmental defect of schizonts of the *Pf*START1 inhibitors at schizonts when they are exposed from ring-stages but this is not significant enough to be an additional mechanism of action of the compounds. It could, however, point to *Pf*START1 being used (but not essential) during schizogony with possibly other START proteins or lipid binding

proteins having more important or overlapping roles at this stage. To better reflect this in the text we have now added the following in the results section:

"To do this, we treated highly synchronous ring-stage parasites with DMSO, M-833 or W-991 and followed them through a cycle of growth, which did not indicate any impact on already developed rings or trophozoites (Fig S5A). We did, however, observe a slight developmental delay of schizonts upon compound treatment at 36 h. Despite this delay, the presence of dysmorphic rings in the M-833 and W-991-treatments was confirmed after 48 h (Fig S5A, 6B)."

We have also altered the FigS5 legend title as follows:

"Supplementary Figure 5. PfSTART1 inhibitors have no effect on ring or trophozoite development..."

-The authors claim that the inhibitors have no activity on the liver stage but the data in Fig S8A is only measuring *P. berghei* sporozoite invasion of HC-04 cells, not liver stage development. The conclusion should be changed to no impact on sporozoite invasion. Additionally, the authors need to show that their compounds are also effective against the *P. berghei* blood stage to be able to interpret this result.

Response: We also agree with this statement. To decipher between invasion and liver stage development, we have changed the results heading to:

"PfSTART1 inhibitors do not prevent sporozoite invasion but block parasite transmission to mosquitoes."

We have also added the following paragraph to the discussion to highlight this point and future work required to test the inhibitors against liver stage development. With respect to the *P. berghei* growth assays, this is a tricky experiment to do because the compounds have a short half-life in mice. W-991 does not reduce *P. berghei* parasitemia at concentrations that are effective against *P. falciparum* and at this stage we do not know if this is because W-991 is unstable in the animal or is not as effective against PbSTART due to structural differences compared to the Pf protein. The mouse studies are described another study under review for the European Journal for Medicinal Chemistry (Nguyen et al. 2024).

*"The activity profile of the PfSTART1 inhibitors was found to be variable in other stages of the lifecycle outside the blood stage with no drug effect against liver stage invasion, stage V gametocyte development and gametogenesis. The lack of activity against liver stage of infection was surprising given that sporozoites grow and replicate in a PV, similar to the asexual blood stages^{60,61}. The apparent inactivity of the PfSTART1 inhibitors could be due to the experimental design where *P. berghei* sporozoites and W-991 were added to human liver*

cells and evaluated only 2 h post invasion. It is possible we did not capture the inhibition of PV formation and further experiments where sporozoites are added to liver cells with the compounds for a longer period may indicate growth inhibition."

Minor Comments:

Comment: -Can an internal population of PfSTART1 be detected by IFA that might correspond with the lower molecular weight PfSTART1 species?

Response: By IFA, we detected internal signal, especially in merozoites. However we did not include our IFA data in the current manuscript as we could not confidently co-localize PfSTART1 with other marker proteins (you can find some of this data on our pre-print: <https://www.biorxiv.org/content/10.1101/2023.11.02.565411v1>). It is possible that PfSTART1_{proc1/2} is packed inside merozoites in preparation for invasion (which would explain the proteinase K assay results). However, we would not be able to tell the difference by IFA between the original and processed forms of PfSTART1 using either HA or PfSTART1 antibodies. More work will be required to fully characterize the different processed forms of PfSTART1 (localization, roles, interacting partners).

Comment: -Is a higher molecular weight band observed outside the cropped area in the blot in Fig S2D that would correspond to a population that did not skip at P2A? If P2A is not skipping efficiently in this fusion, it would strengthen the idea of a reduced functional pool of PfSTART1.

Response: The full (uncropped) western blots have now been provided in the supplementary figures: in the majority of the blots with SLI samples, no band is observed that would correspond to 'unskipped' PfSTART1. The exception being the first replicate of the timecourse (protein extractions along intraerythrocytic development), where a faint band can be detected by the PfSTART1 and the HA antibodies, at around 88 kDa, which likely corresponds to the 'unskipped' protein (PfSTART1 ~54 kDa + 3xHA ~3 kDa + P2A ~2 kDa + NeoR ~29 kDa = 88 kDa). Such a band was detected in our first western blots, but mostly disappeared after cloning the SLI parasite lines. Therefore, we think it is unlikely that 'unskipped' proteins would have played a role in the reduction of a functional pool of PfSTART1.

Comment: In the title to the lattice light sheet microscopy section, "membrane disruption" is perhaps misleading. The data show that differentiation into normal ring morphology does not occur while membrane disruption suggests the membrane has been compromised or lost.

Response: This title of this section has now been changed to:

"PfSTART1 inhibitors block differentiation into ring-stage parasites by preventing PVM expansion."

We also altered the figure legend of our model (Fig 9) to now read:

"The M-833 series blocks conversion of newly invaded merozoites into ring-stage parasites by inhibiting the PfSTART1 protein and reducing expansion of the parasitophorous vacuole membrane (PVM)."

Comment:-For clarity, please indicate that the cyan is mitotracker (which marks the merozoite) in the Fig 6 legend as in the legends to the Supplementary movies.

Response: The legend of Fig 7 (previously Figure 6) has been amended to indicate that the cyan represents MitoTracker.

-What do the marks on the Y axis (sphericity) represent in Fig S7A?

Response: These are grouped examples of the time course for sphericity for each condition and the plot is used to highlight the more consistent sphericity in the drug treated case.

Comment: -SLI-WT PfSTART1 migration is quite different between Fig S2D (~50 kDa) and Fig 7A (~60 kDa – although a second marker is needed to gauge this). Are the marker labels correct? It would be helpful if at least two markers could be included in each cropped blot so that the sizes can be estimated (or include the uncropped blots).

Response: The blots have now been cropped differently to include at least two marker labels (especially for the PfSTART1 and HA antibodies). Uncropped blots have also been provided in the supplementary figures. We hope it is now clearer that PfSTART1-HA migrates to similarly on both blots (~60kDa; new Fig3A and new Fig8A).

Comment: (1)-The proteinase protection experiments are difficult to interpret since there is degradation in both the PBS and SAP +protK lanes and no density quantification was provided to back up the claims (ie, PTEX150 was most strongly degraded in sap (see below)). Also, in the replicate in Fig S9E, there is little to no signal in the PBS -protK lane? This prevents comparison of the level of degradation between the PBS and SAP treatments but isn't commented on.

(2) "PfPTEX150 was most strongly degraded upon addition of saponin indicating it was mainly located in the PV." – but PTEX150 shows even more degradation in the Tx100 treatment than the Saponin treatment?

(3) " PfGBP130 was localised in the RBC as it was degraded in the EqtlI supernatant" - I'm confused by this statement since there is no protK addition to the EqtlI supernatant (left most lane in Fig 7E). Perhaps the authors mean "released" instead of "degraded"?

(4) "whilst PfSTART1proc1 and PfSTART1proc2 could reside within the parasite and protected from proteinase K cleavage." - although I agree the proc 1 band is lost in the Tx100 + protK lane, the proc 2 band is hard to interpret (relative to the Tx100 -protK lane) because of the apparent doublet of degradation products migrating in the same area.

Response: We thank the reviewer for these comments, questions and important feedback regarding this assay. These were difficult assays, which required a lot of optimization and repetition and rarely produced '100% clean' results due to membrane breakage and leakage. Find below the responses to individual questions, and after that, an updated version of the whole paragraph referring to this experiment.

(1) No quantification was done as there are only two replicates, and as densitometry on western blots usually show a lot of variability. We agree with the reviewer's comments regarding degradation occurring in PBS + protK which was not commented on previously: we have changed the text to include this (see below). Regarding the lack of proteins in the PBS - protK in the second replicate, we can only assume the sample was lost possibly due to the gel being damaged, but it only occurred once (over seven blots where such a condition was used).

(2) We agree with the reviewer and the text has been updated.

(3) We thank the reviewer for noticing this mistake: we meant indeed that PfGBP130 is detected in the EqtlI supernatant. The text has been updated accordingly.

(4) We agree with the reviewer, and have added this comment in the text.

The result section referring to the proteinase K assay now reads:

"We then performed a proteinase K protection assay to indicate where PfSTART1 was localised after lysing different membranes as per Fig 8E, +/- proteinase K (Fig 8F, Fig S8I). PfGBP130 was localised in the RBC as it was detected in the EqtlI supernatant. Cytoplasmic PfActin-1 was mostly protected from proteinase K degradation unless all the membranes were lysed in TX-100. PfPTEX150 is a PV protein41, and as such we expected it to be degraded mainly upon addition of proteinase K and saponin. In our assay however, PfPTEX150 was slightly degraded by proteinase K in PBS, further degraded in saponin and completely degraded in TX-100. This suggests that the PVM in our schizonts might have been partially compromised in the PBS treatment. The incomplete degradation of PfPTEX150 in the saponin condition suggests either that the saponin lysis was not complete and some PVM remained unruptured, or that a

fraction of PfPTEX150 remains inside the parasite (newly synthesised, not yet secreted into the PV). A pattern identical to PfPTEX150 was observed for PEXEL-cleaved PfSTART1, suggesting that more of this form of the protein is in the PV than in the parasite. PfSTART1proc1 appears to reside within the parasite as it seems to be protected from proteinase K cleavage unless TX-100 was added. For PfSTART1proc2, a similar conclusion is harder to draw considering the degradation products co-migrate in the same size. Altogether this data corroborates previous work^{12,16,17} that indicates PfSTART1 is most strongly expressed in schizonts, associated with membranes and that more is located in the PV than in the parasite."

Comment: -What are the lower bands partly cropped out in the GBP130 blot in replicate proteinase protection assay in Fig S9E?

Response: The lower bands are due to the blot being previously probed for PfHSP101: the uncropped blot has been provided.

Comment: -Do the changes in the SEC elution profile of recombinant PfSTART1 between WT and N309K (Fig S5A) provide any clue to how this mutation might be impacting the protein?

Response: The major species for PfSTART1 N309K has a retention volume that suggests either the formation of soluble aggregate or higher order oligomers (we have not calibrated the column with molecular weight standards to know the precise void volume). If soluble aggregate, this would suggest the N309K mutation has impacted the stability of the protein, potentially by changing the composition of lipids that can be accommodated within the binding pocket, or through other structural rearrangements. Further investigations are required to uncover the impact of the N309K mutation on PfSTART1.

-A "be" should be inserted before "protected" in the sentence: "...could reside within the parasite and protected from proteinase K cleavage."

Response: Changed. The text now reads: *"appears to reside within the parasite as it seems to be protected from proteinase K cleavage"*

- "Sensitize" should be "sensitivity" in the sentence: "...could interfere with PfSTART1 function and sensitize of SLI-WT parasites to M-833".

Response: Changed. The text now reads: *"could interfere with PfSTART1 function and sensitivity of SLI-WT parasites"*

-Cut "intervals" or change "every" to "at" in the sentence: "...proteins of tightly synchronized wildtype 3D7 schizonts without the HA tag were analysed every 4 h intervals"

Response: Changed. The text now reads: *“proteins of tightly synchronised wildtype 3D7 schizonts without the HA tag were analysed every 4 h”*

Reviewers' Comments:

Reviewer #1:

Remarks to the Author:

Dans et al. and colleagues addressed all the remarks in the revision, so I have no further comments.

Reviewer #2:

Remarks to the Author:

The authors have addressed all my concerns in the revised manuscript. This is a thorough and well conducted study that will be of great interest to the field due to its important new findings regarding novel antimalarial development and the biology of the parasite vacuole.